# TGFβ-induced degradation of TRAF3 in mesenchymal progenitor cells causes age-related osteoporosis

Jinbo Li[1], Akram Ayoub[1], Yan Xiu[1,3], Xiaoxiang Yin[1,4], James O. Sanders[2,5], Addisu Mesfin[2], Lianping Xing[1], Zhenqiang Yao[1] & Brendan F. Boyce[1]

Inflammaging induces osteoporosis by promoting bone destruction and inhibiting bone formation. TRAF3 limits bone destruction by inhibiting RANKL-induced NF-κB signaling in osteoclast precursors. However, the role of TRAF3 in mesenchymal progenitor cells (MPCs) is unknown. Mice with TRAF3 deleted in MPCs develop early onset osteoporosis due to reduced bone formation and enhanced bone destruction. In young mice TRAF3 prevents β-catenin degradation in MPCs and maintains osteoblast formation. However, TRAF3 protein levels decrease in murine and human bone samples during aging when TGFβ1 is released from resorbing bone. TGFβ1 induces degradation of TRAF3 in murine MPCs and inhibits osteoblast formation through GSK-3β-mediated degradation of β-catenin. Thus, TRAF3 positively regulates MPC differentiation into osteoblasts. TRAF3 deletion in MPCs activated NF-κB RelA and RelB to promote RANKL expression and enhance bone destruction. We conclude that pharmacologic stabilization of TRAF3 during aging could treat/prevent age-related osteoporosis by inhibiting bone destruction and promoting bone formation.

---

[1] Department of Pathology and Laboratory Medicine, University of Rochester Medical Center, Rochester, NY 14642, USA. [2] Department of Orthopaedics and Rehabilitation Medicine, University of Rochester Medical Center, Rochester, NY 14642, USA. [3] Present address: Department of Pathology, University of Iowa, Iowa City, IA 52242, USA. [4] Present address: Department of Medical Imaging, Henan University First Affiliated Hospital, 357 Ximen Street, Kaifeng 475001 Henan, P.R. China. [5] Present address: Department of Orthopaedics, University of North Carolina, Chapel Hill, NC 27514, USA. Correspondence and requests for materials should be addressed to Z.Y. (email: Zhenqiang_Yao@urmc.rochester.edu) or to B.F.B. (email: Brendan_Boyce@urmc.rochester.edu)

Osteoporosis is a common disease of aging, characterized by low bone mass, resulting from increased bone resorption and reduced bone formation that lead to fragile bones and increased risk of fractures[1–3]. It is associated with sex steroid deficiency[4] and low-grade chronic inflammation of aging (inflammaging)[5,6]. These are accompanied by increased production of pro-inflammatory cytokines, including TNF, IL-1, and IL-6[5,6], which increase the expression of RANKL, a TNF superfamily member that is expressed by osteoblastic[7] and immune cells[8,9] in bone marrow (BM) and is required for osteoclastogenesis[4,10,11]. RANKL activates NF-κB and other signaling pathways in osteoclast precursors (OCPs) through its receptor, RANK, and the adaptor protein, TNF receptor-associated factor 6 (TRAF6), to induce osteoclastogenesis[10,11].

TRAF3 is also a TNF receptor family adaptor protein, but unlike TRAF6, it limits RANKL- and TNF-induced osteoclastogenesis[12]. It forms a complex in unstimulated B cells with the ubiquitin ligases, cellular inhibitors of apoptosis (cIAP1 and cIAP2), and TRAF2 to constitutively degrade NF-κB-inducing kinase (NIK)[13] and thus it prevents processing of NF-κB p100 to p52 and activation of non-canonical signaling[13]. RANKL induces TRAF3 autophagolysosomal degradation in OCPs to promote osteoclastogenesis[14]. RANKL can also mediate osteoclastogenesis in vitro in the absence of TRAF6[15] by degrading TRAF3, despite reports that TRAF6 expression is required for RANKL-induced osteoclast formation in vitro[16,17]. In addition, mice with TRAF3 conditionally deleted in myeloid progenitor cells (which include OCPs) develop early onset osteoporosis due to increased bone resorption, indicating that TRAF3 expression in these cells limits age-related bone loss[14]. Importantly, prevention of TRAF3 degradation by the autophagolysosome inhibitor, chloroquine, inhibits RANKL-induced osteoclast formation in vitro and prevents PTH-induced osteoclastogenesis and ovariectomy-induced osteoporosis in mice[14], suggesting that inhibition of TRAF3 degradation in OCPs could prevent osteoporosis in humans. However, the role, if any, of TRAF3 in mesenchymal or osteoblast lineage cells has not been investigated.

Osteoblasts are derived from mesenchymal progenitor cells (MPCs), which express Prx1[18,19]. Although Prx1 is expressed predominantly in appendicular bones in embryos[18], it is also expressed in developing vertebrae; and mice with global knockout of Prx1 have defects in the skull, limbs, and vertebrae[18,19]. Prx1Cre mice have been used to conditionally delete genes in MPCs to investigate their roles in skeletogenesis as well as in bone modeling and remodeling[20]. Here, we crossed Prx1Cre with Traf3flox/flox mice to delete TRAF3 in MPCs to determine if TRAF3 has a function in MPCs. These TRAF3 conditional knockout (cKO) mice develop early onset osteoporosis due to a combination of increased bone resorption and decreased bone formation. We found that TRAF3 protein levels are lower in the bone and BM of old than young WT mice and, importantly, in samples of bone from older adults than from children. These findings suggest that prevention of TRAF3 degradation in bone cells could be a novel therapeutic approach to prevent age-related osteoporosis.

## Results

**TRAF3 cKO mice develop early onset osteoporosis.** To determine if TRAF3 has a role in osteoblastic bone formation, we crossed Traf3flox/flox mice[14,21] with Prx1cre mice to generate Traf3f/fPrx1cre (cKO) mice with TRAF3 conditionally deleted in MPCs. The cKO mice appeared normal at birth and later in life, and trabecular bone volume (BV/TV) assessed by micro-CT in tibial metaphyses of 3-month (m)-old cKO mice was similar to that in Traf3f/f (WT) littermates (Fig. 1a). However, BV/TV

values in tibial metaphyses of 9- and 15-m-old cKO mice were significantly lower than in WT mice (Fig. 1a). Similarly, vertebral BV/TV values were normal in 3-m-old cKO mice, but were significantly reduced in 9- and 15-m-old cKO mice compared with their respective WT littermates (Fig. 1b). As is typical during aging, BV/TV values in the tibiae and vertebrae of 15-m-old WT mice were lower than those of younger mice (Fig. 1a, b). We found no differences in the degree of bone loss between male and female cKO mice.

TRAF3 protein levels were lower in samples of femora and vertebrae from 15-m-old than from 3-m-old WT mice (Fig. 1c). They were barely detectable in femoral samples of the 3- and 15-m-old cKO mice, but were detectable at lower levels in vertebral samples from 3-m-old cKO than from WT mice (Fig. 1c), presumably reflecting the presence in these samples of large numbers of hematopoietic cells, which had largely been flushed from the femora. In contrast, TRAF3 levels were almost undetectable in the vertebral samples from the 15-m-old cKO mice (Fig. 1c). This reflects the reduction in TRAF3 levels in whole BM samples that we observed in WT mice during aging (Supplementary Fig. 1a). Although Prx1 also regulates chondroblast development and is expressed in craniofacial bones[20,22,23], we found that long bone growth plates and articular cartilage were histologically normal in young and adult cKO mice (Fig. 1d), which also had normal body length and craniofacial development and no features of osteoarthritis. Thus, TRAF3 does not play a significant role in MPCs in the regulation of skeletal modeling or bone mass during embryonic and postnatal development up to at least 3–4 months after birth.

**Age-related reduction of bone formation in TRAF3 cKO mice.** Very few trabeculae were present in the long bone metaphyses of 9- and 15-m-old cKO mice, reflecting the marked acceleration of age-related bone loss. Thus, we evaluated bone formation in vertebral sections of these mice because they had adequate numbers of trabeculae for analysis. Bone formation rates were normal in 3-m-old cKO mice, but they were significantly lower in 9-m-old cKO mice than in WT littermates (Fig. 1e). Similarly, mean trabecular osteoblast surface values were normal in 3-m-old cKO mice, but were significantly lower in 9-m-old cKO mice than in WT littermates (Fig. 1f). Osteoblast surface and bone formation values in 15-m-old cKO mice were similar to those in WT littermates because these values had decreased in the WT mice by this age (Fig. 1e, f), reflecting the reduction that typically occurs during aging[24]. Consistent with these findings, serum levels of the bone formation marker, osteocalcin, were similar in 3-m-old WT and cKO mice and in 9-m-old WT mice, but they were significantly lower in 9-m-old cKO mice (Fig. 1g). However, values in 15-m-old WT mice were similar to those in 9- and 15-m-old cKO mice because values in the WT mice had decreased by this age, which is typical of aging (Fig. 1g).

**Age-related increase in bone resorption in TRAF3 cKO mice.** We assessed bone resorption parameters in vertebral sections of cKO mice and found that osteoclast numbers and surfaces were normal in 3-m-old mice, but were increased in 9- and 15-m-old cKO mice compared to WT littermates (Fig. 1h). Consistent with this, serum levels of the bone resorption marker, TRACP5b, were normal in 3-m-old cKO mice and were significantly increased in 9- and 15-m-old cKO mice (Fig. 1i). Thus, the early onset osteoporosis in the TRAF3 cKO mice is a result of decreased bone formation and increased bone resorption, which is similar to age-related osteoporosis in WT mice[25] and humans[26]. Although osteoclast number and activity generally are reduced with age in mice due to aging-associated low bone turnover, we found no

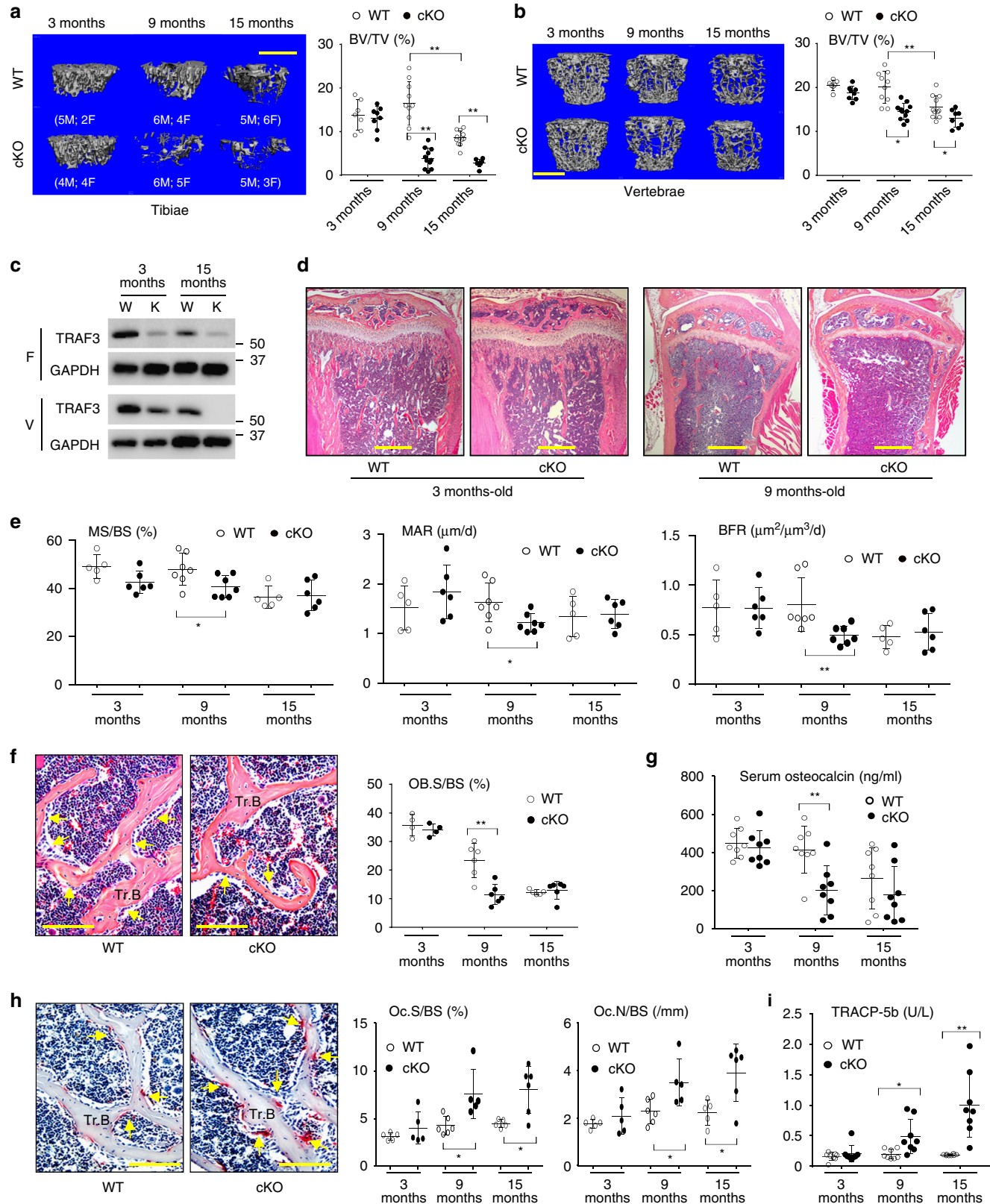

significant differences in osteoclast numbers and surfaces in the vertebral sections or in serum TRACP5b levels among 3-, 9-, and 15- month-old WT mice (Fig. 1h, i). This may reflect the fact that 15-month-old WT mice are still not aged and are equivalent to ~50-year-old humans[27] who still tend to have normal bone turnover marker levels[28].

**Age-related reduction in OB differentiation from cKO MPCs.** To investigate the molecular mechanisms for the reduced bone formation in the cKO mice, we cultured BM cells to expand stromal cells, followed by treatment with osteoblast-inducing medium[29]. Alkaline phosphatase (ALP)-positive colony formation from BM cells from 3-m-old cKO mice was similar to that

**Fig. 1** TRAF3 cKO mice have early onset osteoporosis. **a**, **b** Representative µCT images and bone volume (BV/TV) values in **a** tibiae and **b** L1 vertebrae from litters of 3-, 9-, and 15-month (m)-old WT and TRAF3 conditional knockout (cKO) mice, including both males and females, as listed in the figure. Mean ± SD (3-m-old WT ($n = 7$) and cKO ($n = 8$), 9-m-old WT ($n = 10$) and cKO ($n = 11$), 15-m-old WT ($n = 11$) and cKO ($n = 8$) biologically independent mice; *$p < 0.05$, **$p < 0.01$). Scale bar, 1 mm. **c** Western blot (WB) of TRAF3/GAPDH in lysates of femoral metaphyses (F; upper panel) and vertebrae (V; lower panel) from 3- and 15-m-old WT and cKO mice. **d** Representative H&E-stained sections of proximal tibiae of 3- and 9-m-old WT and TRAF3 cKO mice. Scale bar, 400 µm. **e** Mineralizing surface (MS/BS), mineral apposition rate (MAR), and bone formation rate (BFR) analyzed in calcein double-labeled plastic sections of L1 vertebrae from the mice in (**a**). Mean ± SD (3-m-old WT ($n = 5$) and cKO ($n = 6$), 9-m-old WT ($n = 7$) and cKO ($n = 7$), 15-m-old WT ($n = 5$) and cKO ($n = 6$) biologically independent samples; *$p < 0.05$, **$p < 0.01$). **f** Representative H&E-stained images of vertebral sections from 9-m-old WT and cKO mice showing osteoblasts (yellow arrows) on the trabecular bone (Tr.B) surfaces, and histomorphometric analysis of osteoblast surfaces (OB.S/BS) from the mice in (**a**). Mean ± SD (3-m group: $n = 4$, other groups: $n = 6$ biologically independent samples; **$p < 0.01$). Scale bar, 100 µm. **g** Serum osteocalcin values tested by ELISA from the 3-, 9-, and 15-m-old WT and cKO mice in (**a**). Mean ± SD ($n = 8$ biologically independent samples; **$p < 0.01$). **h** Representative images of TRAP-stained vertebral sections from 9-m-old WT and cKO mice, and histomorphometric analysis of osteoclast numbers (Oc.N) and surfaces (Oc.S) from the mice in (**a**). Mean ± SD (3-m group: $n = 5$, other groups: $n = 6$ biologically independent samples; *$p < 0.05$). Scale bar, 100 µm. **i** Serum TRACP-5b levels tested by ELISA from the mice in (**a**). Mean ± SD ($n = 8$ biologically independent samples; *$p < 0.05$, **$p < 0.01$). All analyses done using one-way ANOVA with Tukey's post-hoc test

of cells from WT mice (Fig. 2a, b). In contrast, ALP⁺ colony formation from BM cells from 9-m-old cKO mice was significantly reduced (Fig. 2a, b). However, the total area of ALP⁺ and ALP⁻ cell colonies formed from BM cells from 9-m-old cKO and WT BM cells was similar (Fig. 2b). Although ALP⁺ osteoblast differentiation from bone-derived MPCs (BdMPCs)[29] from 3-m-old cKO mice was similar to that from WT littermate mice, it was significantly lower in cells from 9-m-old cKO mice than from WT littermates (Fig. 2c). In contrast, BdMPCs from 3- and 9-m-old cKO mice had faster growth rates than those of their respective littermates (Fig. 2d), which was supported by the findings that cKO BdMPCs had a higher S-phase cell population than control mice (Fig. 2e). The total area of cells from BM and BdMPCs was similar between cKO and WT in Fig. 2a, c because the cKO cells had reached confluence at this stage. These findings suggest that the reduced bone formation during aging in the mice with TRAF3 deleted in MPCs is mainly the result of reduced osteoblast differentiation and not reduced MPC proliferation and could be due to exposure to increasing concentrations of a factor(s), such as TGFβ, during aging.

**TGFβ limits OB differentiation of MPCs from 9-m-old cKO mice.** TGFβ1, one of the most abundant non-collagenous proteins in bone matrix, is released during bone resorption and activated by the acid environment under osteoclasts in resorption lacunae[30]. Although TGFβ1 couples bone formation to resorption by recruiting MPCs[30], it actually inhibits osteoblast differentiation from these cells[31–33], and thus could mediate the inhibitory effects we observed in the cKO cells. We found that the concentrations of total (Fig. 2f) and the active form (Fig. 2g) of TGFβ1 in serum were significantly increased in 9-m-old cKO mice, but not in 3-m-old cKO mice, compared to their respective WT littermates. However, BdMPCs from 3- and 9-m-old cKO mice expressed levels of TGFβ1 similar to those in cells from their respective littermates (Supplementary Fig. 2), suggesting that these cells were not the source of the increased serum TGFβ1 levels. To investigate if the increased active TGFβ1 in the older cKO mice is responsible for the reduction in osteoblast differentiation, we treated BdMPCs[29] from 3- and 9-m-old WT and cKO mice with TGFβ1 for 7 days followed by induction of osteoblast differentiation without TGFβ1. The area of ALP⁺ osteoblasts formed from BdMPCs from 3-m-old cKO mice was similar to that from cells from WT littermates (Fig. 2h), expressed either as mm² (Fig. 2i) or percentage of the well area (Fig. 2j), but it was significantly lower in cells from 9-m-old cKO than WT mice (Fig. 2k–m). Interestingly, MPCs from both WT and cKO mice pre-treated with TGFβ1 were unable to differentiate into osteoblasts (Fig. 2h–m).

**TGFβ1 degrades TRAF3 in bone during aging.** TRAF3 protein levels were significantly lower in samples of tibial metaphyses, femoral BM, and femoral cortical bone from 18- than from 3-m-old WT mice (Fig. 3a and Supplementary Fig. 1a & b). Importantly, they were significantly lower in samples of vertebral spinous processes removed surgically from older otherwise healthy adults than in those from children (Fig. 3b and Supplementary Fig. 1c), with no differences detected between males and females. In addition, using immunofluorescence we found that TRAF3 was expressed strongly in osteocalcin-expressing osteoblasts on trabecular and endosteal bone surfaces in decalcified vertebral samples from young adult mice (Fig. 3c). The numbers of these double-positive cells on bone surfaces were significantly lower in samples from 18-m-old WT mice (Fig. 3c). In addition, there were significantly fewer TRAF3⁺ cells in the BM of the old mice (Fig. 3c). These findings are consistent with a recent report that TRAF3 levels are lower in circulating blood monocytes from older (>65-years-old) than from younger humans (20–30-years-old)[34].

Since RANKL induces TRAF3 degradation in myeloid progenitors to promote osteoclastogenesis[14], we next examined if factors that regulate osteoblast differentiation might degrade TRAF3 in MPCs. We found that TGFβ1, but not TNF, PTH, or BMP2, reduced TRAF3 protein levels in WT mouse MPCs (Fig. 3d). The levels of total TGFβ1 protein in bone samples (Fig. 3e) and of active TGFβ1 protein in BM, but not in serum (Fig. 3f), from 19-m-old WT mice were significantly higher than those in young (2.5-m-old) mice. Consistent with this, levels of the active form of TGFβ1 in vertebral samples from older adults (55–87-years-old) were higher than those in samples from children, although there was no difference in total TGFβ1 levels (Fig. 3g).

**TGFβ1 induces TRAF3 ubiquitination and lysosomal degradation.** TGFβ1 increased TRAF3 ubiquitination in MPCs (Fig. 3h). TRAF3 is degraded in B cells by cIAP1 and cIAP2[13]. We found that both cIAP1/2 and TRAF3 bound to the TGFβ receptor I (TGFβRI; Fig. 3i), and that cIAP1/2 bound to TRAF3 in MPCs (Fig. 3i). We also confirmed that TGFβRI was present in these lysates using immunoprecipitation and WB (Fig. 3i, bottom). TGFβ1 increased binding of TRAF3 and cIAP2 to TGFβRI and of cIAP2 to TRAF3 (Fig. 3i). Treatment of these cells with the IAP inhibitor, AT406, which degrades cIAP1 and cIAP2[35], reduced these interactions and almost completely abrogated binding of TRAF3 to TGFβRI (Fig. 3i), suggesting that TRAF3 binding to TGFβRI requires cIAP1/2. Previous studies have reported that phosphorylation of TGFβR1 is associated with its ubiquitination[36]. Our findings suggest that cIAP1/2 and TRAF3

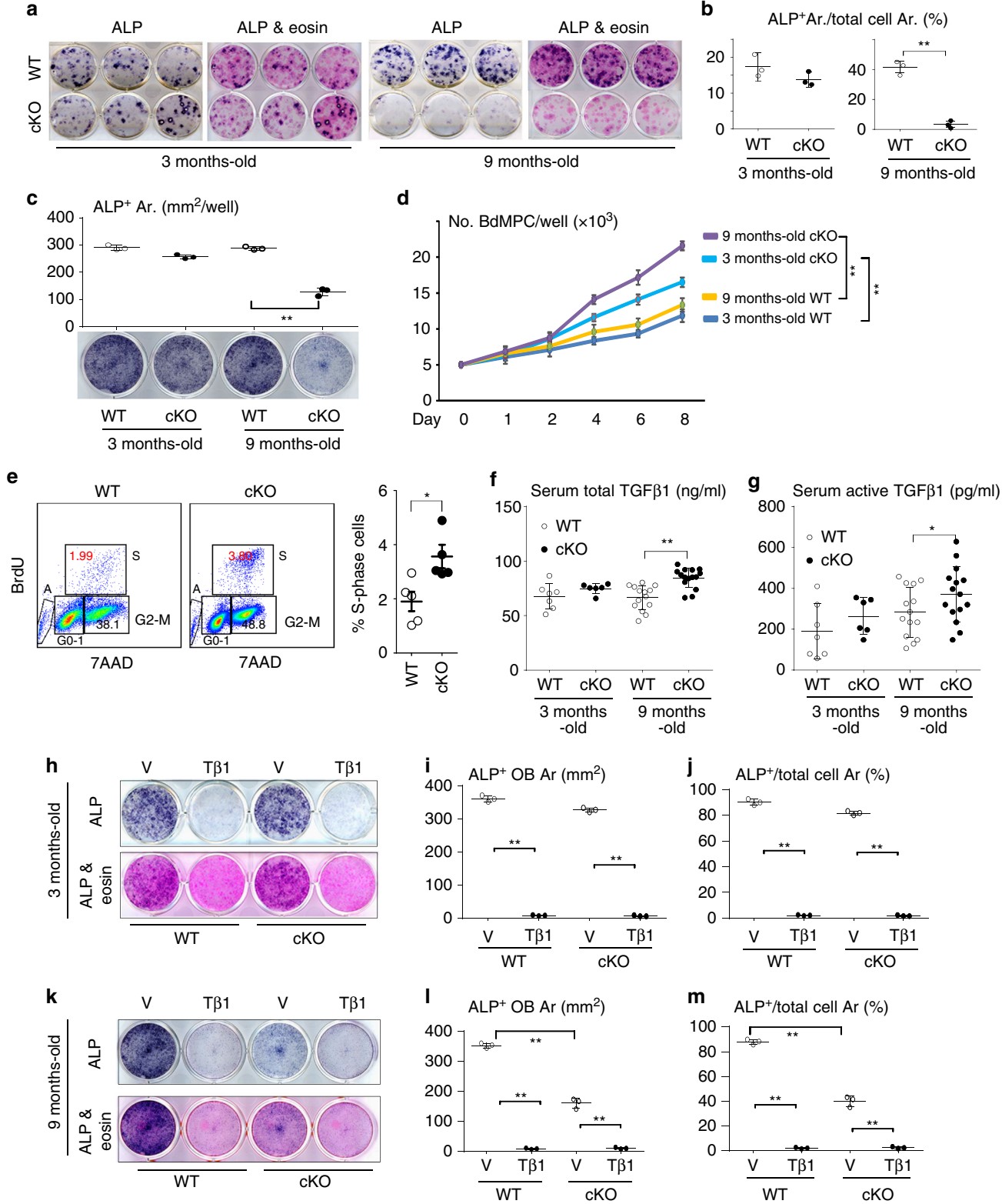

form a complex on the TGFβRI that results in TRAF3 ubiquitination and degradation. Interestingly, AT406 almost completely prevented TGFβ1 induction of cIAP binding to TRAF3 (Fig. 3i) since it markedly reduced cIAP protein levels in MPCs (Fig. 3j). However, AT406 did not completely prevent TGFβ1-induced inhibition of osteoblast differentiation (Fig. 3k), suggesting that an IAP-independent pathway may also be involved in TGFβ1-induced inhibition of osteoblast differentiation.

Treatment of BdMPCs from 3-m-old WT mice with TGFβ1 plus the lysosome inhibitor, chloroquine, significantly increased ubiquitinated TRAF3 (Fig. 3h), suggesting that chloroquine prevented TRAF3 degradation. The proteasome inhibitor, MG-132, also increased the amount of ubiquitinated TRAF3, but to a much lesser extent (Fig. 3h), suggesting that TRAF3 is degraded predominantly by lysosomes in MPCs, similar to its degradation in OCPs[14]. Consistent with this, TGFβ1 increased the area of

**Fig. 2** Impaired OB differentiation from cKO MPCs primed by TGFβ1. **a, b** Osteoblastic cells derived from BM cells from 3- and 9-m-old WT and TRAF3 cKO mice stained for alkaline phosphatase (ALP) activity (**a**). Some wells were counter-stained with eosin to assess ALP+ and total cell area (Ar.) and the % ALP+/total cell area (**b**). Mean ± SD ($n = 3$ biologically independent samples; **$p < 0.01$). **c** Osteoblastic cells derived from bone-derived MPCs (BdMPCs) from 3- and 9-m-old WT and cKO mice stained for ALP activity to quantify ALP+ cells. Mean ± SD ($n = 3$ biologically independent samples; **$p < 0.01$). **d** Numbers of cells derived from BdMPCs from 3- and 9-m-old WT and cKO mice cultured for 1–8 days in 6-well plates. Mean ± SD ($n = 3$ biologically independent samples; **$p < 0.01$). **e** Cell cycle status of previously starved BdMPCs from 3-m-old WT and cKO mice analyzed by flow cytometry. Mean ± SD ($n = 5$ biologically independent samples; *$p < 0.05$). **f, g** Total (**f**) and active (**g**) TGFβ1 levels in serum from 3- and 9-m-old WT and cKO mice tested by ELISA. Mean ± SD; from left to right: 3-m-old WT ($n = 7$) and cKO ($n = 6$), 9-m-old WT ($n = 14$) and cKO ($n = 15$ biologically independent samples; *$p < 0.05$, **$p < 0.01$). **h–j** BdMPCs generated from 3-m-old WT and cKO mice pre-treated with vehicle (V) or TGFβ1 (Tβ1; 1 ng/ml) for 7 days followed by treatment for OB differentiation for 7 days without TGFβ1. **i** ALP+ cell area and **j** ALP+ area % in total area. Mean ± SD ($n = 3$ biologically independent samples; **$p < 0.01$). **k–m** Data similar to those in (**h–j**) for 9-m-old WT and cKO mice. Mean ± SD ($n = 3$ biologically independent samples; **$p < 0.01$). Analyses in (**b**) and (**e**) done using unpaired Student's $t$ test; all others done using one-way ANOVA with Tukey's post-hoc test. All in vitro experiments were repeated twice with similar results

co-localization of TRAF3 with the lysosome marker, LAMP2, in the cytoplasm of MPCs (Fig. 3l). Importantly, chloroquine inhibited TGFβ1-induced TRAF3 degradation (Fig. 3m) and dose-dependently prevented the inhibition of osteoblast differentiation induced by TGFβ1 (Fig. 3n) in WT cells. Chloroquine also reduced the inhibition of osteoblast differentiation induced by TGFβ1 in cKO cells, but only at the highest concentration tested (3000 nM), which supports lysosomal degradation of TRAF3 being a major, but not the only mechanism whereby TGFβ inhibits OB formation.

**TRAF3 maintains β-catenin activity in MPCs.** β-Catenin signaling regulates many crucial biological processes, including MPC fate and osteoblast differentiation[37], while inactivation of β-catenin prevents MPC differentiation into osteoblasts[38]. Total β-catenin protein levels were lower in bone samples from young and adult TRAF3 cKO mice than from their respective WT controls (Fig. 4a). Consistent with this, treatment of WT MPCs with TGFβ1 for 2–4 days markedly reduced total β-catenin protein levels, associated with reduced TRAF3 protein levels (Fig. 4b). We also assessed the expression of β-catenin in nuclei of WT and cKO MPCs derived from neonatal calvarial cells infected with a control pMX-GFP or pMX-TRAF3 retrovirus using immunofluorescence (Fig. 4c). We found that the basal level of β-catenin, assessed as the area of nuclear staining, was lower in the cells from cKO than from WT mice (Fig. 4c), suggesting that activation of β-catenin is TRAF3-dependent. TGFβ1 reduced the area of β-catenin staining in nuclei in MPCs from both control virus-infected WT and cKO cells (Fig. 4c), suggesting that TGFβ1 also regulates β-catenin activation through a TRAF3-independent mechanism. Indeed, over-expression of TRAF3 blocked the reduction of nuclear β-catenin staining induced by TGFβ1 in WT and cKO cells (Fig. 4c), further confirming that activation of β-catenin is through TRAF3. We confirmed that TRAF3 retrovirus infection of calvarial cells caused over-expression of TRAF3 (Fig. 4d). However, over-expression of TRAF3 in unstimulated cKO cells did not restore the degree of β-catenin translocation to that observed in unstimulated WT cells, probably because GSK-3β is constitutively activated in unstimulated cells to degrade β-catenin, and over-expression of TRAF3 does not stimulate OB differentiation via β-catenin.

**TRAF3 limits GSK3β activity to prevent β-catenin degradation.** β-Catenin is phosphorylated and degraded by GSK3β[39]. In general, GSK3β activity is increased by Tyr216 phosphorylation and reduced by Ser9 phosphorylation[40]. GSK3β Tyr216 phosphorylation was increased and Ser9 phosphorylation was reduced in TRAF3 cKO MPCs (Fig. 4e), with no difference in total GSK3β levels. We generated GSK3β plasmids in which Tyr216 and Ser9

were mutated such that they could not be phosphorylated (Y216m and S9m, respectively), and infected MPCs with them or WT GSK3β-expression vectors. Over-expression of these GSK3β constructs was confirmed by increased HA levels in the MPCs (Fig. 4f). Over-expression of mutated Tyr216-GSK3β (Y216m; as a dominant negative GSK3β) markedly reduced Tyr216 phosphorylation, associated with increased β-catenin in vehicle-treated cells. It also inhibited TGFβ1-induced degradation of β-catenin (Fig. 4f). In contrast, over-expression of mutated Ser9-GSK3β (S9m) had little effect on total β-catenin levels in either vehicle- or TGFβ1-treated cells compared to those in WT-GSK3β-infected cells (Fig. 4f). These findings suggest that TGFβ1-induced Tyr216 phosphorylation activates GSK3β to cause β-catenin degradation.

Consistent with the above findings, over-expression of TRAF3 in vehicle-treated cells decreased Tyr216 and increased Ser9 phosphorylation of GSK3β, associated with decreased phospho-β-catenin and increased total β-catenin (Fig. 4g). Over-expression of TRAF3 also reduced TGFβ1 induction of phospho-Tyr216 and phospho-β-catenin, and thus partly reversed TGFβ1-induced reduction of total β-catenin levels (Fig. 4g). Importantly, GSK3β directly associated with TGFβRI (Fig. 4h), and over-expression of TRAF3 reduced TGFβ1-induced binding of GSK3β to TGFβRI. In particular, it reduced binding of Tyr216-phosphorylated GSK3β to the TGFβRI (Fig. 4h, lower panel). In addition, the GSK3β inhibitor, SB-216763[41] prevented TGFβ1-induced inhibition of OB differentiation from[20] WT BdMPCs (Fig. 4i, j), and reduced the increase in protein levels of p-GSK3β (Tyr216) (Fig. 4k, l). The inhibitor also prevented TGFβ1-induced inhibition of OB differentiation from cKO cells (Fig. 4i, j) and the increase in p-GSK3β (Tyr216) in cKO cells (Fig. 4k, l), but this required higher concentrations of the inhibitor, reflecting the higher level of p-GSK3β (Tyr216) in these cells (Fig. 4k, l). In addition, over-expression of TRAF3 in WT MPCs increased the area of ALP+ cells (Fig. 4m, n).

TGFβ1 receptor signaling also regulates osteoblast differentiation by phosphorylating regulatory Smads, including Smad1, 2, 3, 5, and 9, which can form complexes with Smad4 to regulate target gene expression[42]. RelB, a non-canonical NF-κB signaling protein regulated by TRAF3, represses TGFβ target gene expression by binding to the Smad2, 3, and 4 promoters in HEK 293T cells, and repression of these genes can be rescued by inhibition of Smad4[43]. Therefore, we examined canonical TGFβ signaling in TRAF3 cKO MPCs and found 1-fold enhanced basal and TGFβ1-induced phosphorylation of Smad2 and 3 (Supplementary Fig. 3a & b), with normal levels of total Smad2 and 3.

**TRAF3 limits RANKL expression by MPCs.** Increased bone resorption in 9-m-old cKO mice was unexpected (Fig. 1h, i) and

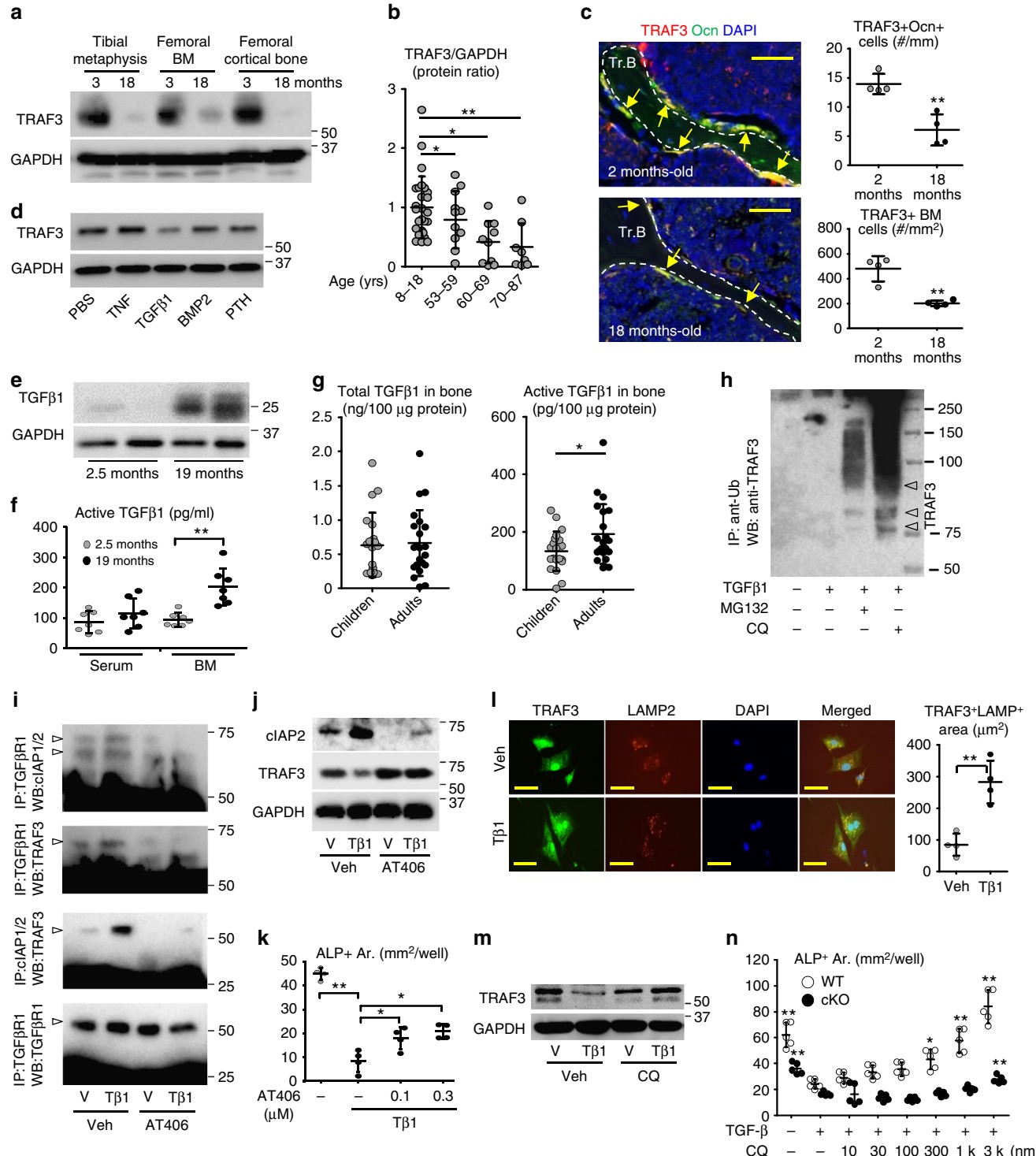

led us to postulate that TRAF3 might negatively regulate expression of RANKL in osteoblastic cells. We co-cultured calvarial pre-osteoblasts with spleen cells from newborn mice in a standard co-culture assay to assess their osteoblastic osteoclastogenic potential[44]. We found that cKO pre-osteoblasts induced significantly more osteoclasts from either cKO or WT spleen cells than WT calvarial cells (Fig. 5a, b), which was blocked by addition of RANK:Fc[45] (Fig. 5b), indicating that TRAF3-deficient MPCs enhance osteoclast differentiation through RANKL. Indeed, levels of RANKL protein (Fig. 5c, d) and mRNA (Fig. 5e) in tibial metaphyseal bone were significantly higher in 3- and

9-m-old cKO than WT MPCs. However, levels of osteoprotegerin (OPG), a RANKL decoy receptor that inhibits RANKL-induced osteoclast formation, were also elevated in samples from 3-m-old cKO mice (Fig. 5c), resulting in a relatively normal RANKL/OPG ratio in these mice (Fig. 5d). Importantly, this ratio was increased in samples from 9-m-old cKO mice (Fig. 5d).

**Activated NF-κB RelA and RelB promote RANKL expression by cKO MPCs.** TRAF3, TRAF2, and cIAPs degrade NIK[46] and negatively regulate lymphotoxin β receptor-mediated canonical and non-canonical NF-κB activation[47]. TGFβ1 markedly

**Fig. 3** Age-related TGFβ activation promotes TRAF3 degradation. **a** TRAF3 and GAPDH WBs of tibial metaphyses, femoral BM and cortical bone from 3- and 18-m-old *C57BL/6* mice. **b** Densitometry of TRAF3 WBs of bones from patients. Mean ± SD; 8–18-years (n = 26), 53–59-years (n = 11), 60–69-years (n = 10), 70–87-years (n = 8 biologically independent samples; *p < 0.05, **p < 0.01). **c** TRAF3 and osteocalcin (Ocn) immunostained vertebral sections. TRAF3[+]/Ocn[+] osteoblasts (yellow arrows) on trabecular bone (Tr.B) surfaces and TRAF3[+] hematopoietic cells in BM. Mean ± SD (n = 4 biologically independent samples; **p < 0.01). Scale bar, 50 μm. **d** TRAF3 and GAPDH WBs of BdMPCs treated with PBS, TNF (20 ng/ml), TGFβ1 (1 ng/ml), BMP2 (100 ng/ml), or PTH (80 ng/ml) for 8 h. **e** WB of TGFβ1 in tibial metaphyseal lysates. **f** Active TGFβ1 levels in serum and BM from 2.5- and 19-m-old *C57BL/6* mice. Mean ± SD (n = 7 biologically independent samples; **p < 0.01). **g** Total and active TGFβ1 levels in vertebral lysates. Mean ± SD (children (8–18-years) n = 22; and adults (55–87-years) n = 20 biologically independent samples; *p < 0.05). **h** BdMPCs treated with vehicle or TGFβ1 +/−chloroquine (100 μM) or MG132 (20 μM) for 8 h. IP using anti-Ub Ab and WB with TRAF3 Ab. Likely mono- and poly-ubiquitinated TRAF3 (lower and upper arrowheads, respectively). **i** Calvarial pre-OBs treated with vehicle or TGFβ1+/−300 nM AT406 for 8 h. IP with anti-TGFβRI Ab and WB with cIAP1/2, TRAF3 and TGFβRI Abs, or IP with anti-cIAP1/2 Ab and WB with TRAF3 Ab. **j** WB of cIAP2, TRAF3, and GAPDH in cells in (**i**). **k** Areas of ALP+ cells from BdMPCs treated with TGFβ1+/−AT406 for 5 days. Mean ± SD (n = 4 biologically independent samples; *p < 0.05, **p < 0.01). **l** IF and area of TRAF3 and LAMP2 co-localization in BdMPCs treated with vehicle or TGFβ1 plus chloroquine for 8 h. Mean ± SD (n = 4 biologically independent samples; **p < 0.01). Scale bar, 20 μm. **m** WB of TRAF3 and GAPDH in BdMPCs treated with vehicle or TGFβ1+/−chloroquine for 8 h. **n** Areas of ALP+ cells from WT and cKO BdMPCs treated with vehicle or TGFβ1+/−chloroquine for 5 days. Mean ± SD (n = 5 biologically independent samples; *p < 0.05, **p < 0.01 vs. TGFβ1 alone. Analyses in (**c, g, l**) done using unpaired Student's *t* test and in (**b, f, k, n**) using one-way ANOVA with Tukey's post-hoc test. All in vitro experiments repeated twice with similar results. Tβ1: TGFβ1 (1 ng/ml)

increased nuclear translocation of RelA and RelB in TRAF3 cKO MPCs (Fig. 5f), suggesting that TRAF3 might regulate RANKL expression in MPCs through them. We detected low levels of RelA and RelB in nuclear fractions of untreated WT MPCs (Fig. 5f). This is consistent with a low level of NF-κB signaling being required to maintain cell survival even in static conditions and massive TNF-induced necrosis of liver cells causing death of *RelA*[−/−] mice in utero[48]. For this reason cells are often starved to clear NF-κB protein from nuclei to facilitate detection of small changes in their translocation[49]. TGFβ1 markedly increased levels of nuclear RelA and RelB in the MPCs (Fig. 5f). We identified three κB binding sites in the distal region of the murine RANKL promoter (Fig. 5g). ChIP assays showed RelA (Fig. 5h) and RelB (Fig. 5i) binding to these κB sites, which was markedly higher in cKO MPCs (Fig. 5h–j). Consistent with this, over-expression of either RelA (Fig. 6a) or RelB (Fig. 6b) in WT mouse calvaria-derived MPCs increased RANKL mRNA expression (Fig. 6c). However, the level of soluble RANKL in the culture medium was not increased (Fig. 6d). Notably, over-expression of either RelA or RelB increased membrane-bound RANKL protein levels (Fig. 6e), consistent with membrane-bound RANKL in osteoblastic cells regulating osteoclast differentiation[50].

**Perilacunar/canalicular remodeling is normal in TRAF3 cKO mice.** TGFβ1 signaling in osteocytes regulates perilacunar/canalicular remodeling[51], which is reduced in osteocyte-specific TGFβRII conditional knockout mice[51], associated with reduced osteoclast numbers and expression of the resorption-related genes, *Acp5, cathepsin K, MMP2, 13* and *14*, and slightly increased bone mass. Thus, the increase in levels of total TGFβ1 in old WT mouse tibiae or in active TGFβ1 in BM in old mice and in adult human bone samples might increase perilacunar remodeling. We did not observe changes in mRNA levels of *Acp5, MMP2, MMP13, MMP14,* or *CTSK* in 15-m-old cKO mice (Supplementary Fig. 4a), in protein expression levels of MMP13 or CTSK in sections of bones of 3-, 9-, or 15-m-old cKO mice, assessed using immunohistochemistry (Supplementary Fig. 4b–d), or in osteocyte lacunar area (Supplementary Fig. 4e), consistent with increased TGFβ1 in bone matrix of old mice being inactive and TGFβ1 activated in resorption lacunae by acid produced by osteoclasts likely not acting on osteocytes in the bone matrix.

## Discussion

This is the first report of an important regulatory role for TRAF3 in MPCs in which it maintains their differentiation into

osteoblasts and restrains their expression of RANKL to limit osteoclastogenesis during aging. In young mice, TRAF3 limits TGFβ1-induced GSK3β activation (through Tyr216 phosphorylation) and degradation of β-catenin in MPCs, which allows β-catenin to maintain osteoblast differentiation and induce OPG expression[52], which limits bone destruction. In contrast, as illustrated in Fig. 7, during aging, increased cytokine production in response to low-grade chronic inflammation[5,6] increases RANKL expression by osteoblastic[7] and immune cells[8,9]. This leads to RANKL-induced TRAF3 degradation in OCPs[14] and thus increased bone resorption through NF-κB signaling[12,14]. As a result, TGFβ is released from bone matrix and induces TRAF3 ubiquitination through cIAPs and subsequent lysosomal degradation in MPCs. Consequently, (1) GSK3β is activated to degrade β-catenin, resulting in inhibition of osteoblast differentiation and reduced expression of OPG[52]; and (2) RelA and RelB induce RANKL expression to further enhance osteoclastogenesis and a self-amplifying cycle of bone destruction, TGFβ release, TRAF3 degradation, and NF-κB activation.

Importantly, the above model (and loss of this protective effect of TRAF3 in the cKO mice, with associated osteoporosis) is supported by significantly lower TRAF3 protein levels in bone from older adults than from children and in bone and BM from old than from young mice, associated with increased levels of active TGFβ1 in the BM of older mice and in the bone of older humans. A caveat with the human bone samples we used is that degenerative spinal conditions in older patients could have led to lower TRAF3 levels, particularly because degenerative arthritis is more common in aged humans[53]. However, unlike vertebral bodies, the spinous processes we collected are not subject to articular wear and tear, a common cause of osteoarthritis[54]. Thus, low TRAF3 levels in older subjects most likely reflect the effects of age-related osteoporosis and not osteoarthritis. These findings support our conclusion that the reduction of TRAF3 in hematopoietic cells, including OCPs, and in MPCs during aging results in enhanced bone resorption and reduced bone formation, respectively (Fig. 7).

The normal bone mass in 3–4-m-old cKO mice is intriguing, given that 9-m-old mice are osteoporotic. We attribute this to two functional changes in MPCs as a consequence of TRAF3 deficiency. First, although RANKL expression by MPCs is increased in 3-m-old cKO mice, OPG is also increased to the same extent because TRAF3 is unavailable to limit β-catenin-induced OPG expression[52] and this prevents bone loss. In contrast, RANKL levels are higher than those of OPG in older cKO mice (Fig. 5d), and this drives increased bone resorption. To our knowledge, this

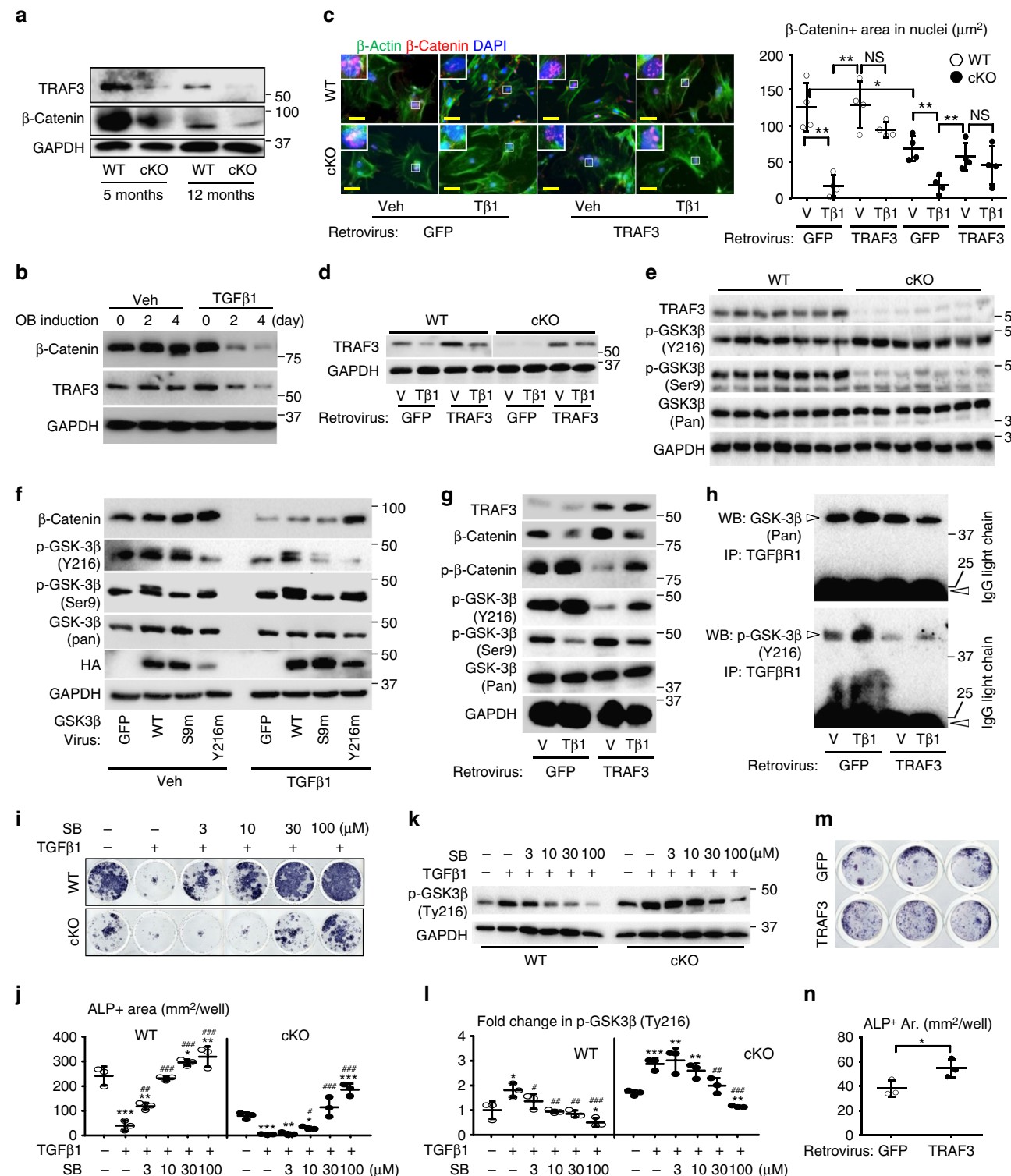

is the first report of RANKL expression being regulated directly by RelA and RelB in any cell type. It identifies another mechanism whereby NF-κB negatively regulates bone mass in addition to inducing osteoclast formation[10,11] and inhibiting osteoblast formation by down-regulating Fra-1[55] and Runx2[29] expression in osteoblast precursors. Activation of the RANKL promoter is complicated because RANKL transcription can be activated by binding of 1,25(OH)$_2$VitD$_3$[56] and PTH-induced CREB[57] to both proximal and distal elements of its promoter in osteoblastic cells and by c-Fos at the distal D5 enhancer in

activated T cells[8]. These findings overall suggest that the distal elements induce RANKL activation[56]. NF-κB also interacts with the transcriptional start site and the D5 enhancer near the Vitamin D response elements[58]. However, we did not find any κB binding sites within 3 kb from the murine RANKL promoter coding start site, but found three between 3 and 5.5 kb. Importantly, RelA and RelB bound functionally to each of these sites.

Second, cKO MPCs have a complex phenotype: increased proliferation in both young and older cKO mice, but their differentiation into osteoblasts is normal in young and impaired in

**Fig. 4** Reduction in TRAF3 activates GSK-3β to impair β-catenin signaling. **a** WB of TRAF3 and β-catenin in long bones. **b** WB of TRAF3, β-catenin, and GAPDH in WT BdMPCs induced for OB differentiation with vehicle or TGFβ1. **c** IF and quantification of nuclear β-catenin and β-actin (cytoskeleton) in WT and cKO calvarial pre-OBs treated with vehicle or TGFβ1 for 48 h following pMX-GFP control or pMX-TRAF3 retrovirus infection. Mean ± SD ($n = 4$ biologically independent samples; *$p < 0.05$, **$p < 0.01$). Scale bar, 20 μm. **d** WB of TRAF3 and GAPDH in cells as in (**c**). **e** WB of TRAF3, phospho-GSK-3β (Tyr216/Ser9), and total GSK-3β in cortical bones from 12-m-old WT and TRAF3 cKO mice. **f** WB of β-catenin, phospho-GSK-3β (Ser9/Tyr216), total GSK-3β, HA, and GAPDH in WT calvarial pre-OBs treated with vehicle or TGFβ1 for 48 h following lentivirus infection with GFP, or HA-tagged WT-, Ser9-mutated (S9m) or Tyr216-mutated (Y216m) GSK-3β. **g** Protein levels of TRAF3, phospho-β-catenin, β-catenin, phospho-GSK-3β (Ser9/Tyr216), and total GSK-3β tested in WT calvarial pre-OBs treated with vehicle or TGFβ1 for 8 h following pMX-GFP or -TRAF3 retrovirus infection. **h** Cell lysates from (**g**) immunoprecipitated using anti-TGFβRI Ab followed by WB of GSK-3β and phospho-GSK-3β (Tyr216). **i** BdMPCs from 4-m-old WT and cKO mice treated with TGFβ1+/−GSK-3β inhibitor, SB-216763, for 7 days and stained for ALP activity. **j** Quantification of ALP+ cell areas. Mean ± SD ($n = 3$ biologically independent samples; *$p < 0.05$, **$p < 0.01$, ***$p < 0.001$ vs. vehicle; #$p < 0.05$; ##$p < 0.01$; ###$p < 0.001$ vs. TGFβ1 alone; one-way ANOVA with Tukey's post-hoc test). **k, l** WB of phospho-GSK-3β (Tyr216) in WT and cKO BdMPCs treated with TGFβ1 plus SB-216763 (**k**), and densitometry analysis was performed (**l**). Mean ± SD ($n = 3$ biologically independent samples; *$p < 0.05$, **$p < 0.01$, ***$p < 0.001$ vs. vehicle; #$p < 0.05$; ##$p < 0.01$; ###$p < 0.001$ vs. TGFβ1 alone;). **m** pMX-GFP or -TRAF3 retrovirus-infected cells induced for OB differentiation for 5 days in 48-well plates and stained for ALP activity. **n** Quantification of ALP+ cell areas. Mean ± SD ($n = 3$ biologically independent samples; *$p < 0.05$; unpaired Student's $t$ test). All other analyses done using one-way ANOVA with Tukey's post-hoc test. All experiments were repeated twice with similar results. Tβ1: TGFβ1 (1 ng/ml)

older mice, associated with increased active TGFβ1 in serum in 9-m-old cKO mice. In addition, total TGFβ1 levels are increased in the metaphyseal bone of 19-m-old WT mice, which also have increased active TGFβ1 levels in their BM. TGFβ1 promotes MPC proliferation, but it also inhibits their differentiation[31–33]. Thus, we propose that the increased MPC proliferation in young cKO mice maintains a sufficient pool of cells to differentiate into osteoblasts and maintain bone mass, but increased release of TGFβ1 as the mice age limits MPC differentiation.

This is the first report linking TRAF3 to TGFβ signaling in any cell type and points to possible important negative regulatory roles in TGFβ signaling in other cell types. TGFβRI indirectly associated with TRAF3 (Fig. 3j) by recruiting cIAP1 and 2, which resulted in TRAF3 degradation followed by inactivation of β-catenin. However, our findings also show that pre-treatment with TGFβ1 prevents osteoblast differentiation from both WT and TRAF3 cKO MPCs, suggesting that TGFβ1 also inhibits bone formation by a TRAF3-independent mechanism.

We found that basal and TGFβ1-stimulated levels of p-Smad 2 and 3 were mildly increased in MPCs from the cKO mice, suggesting that TRAF3 is also involved in canonical TGFβ1 signaling in these cells. Interestingly, loss of expression of Smad4 in MPCs causes stunted growth and spontaneous fractures in mice, associated with increased MPC proliferation[59,60], which is also increased in TRAF3 cKO MPCs. However, this Smad4 cKO phenotype is much more severe than that in our cKO mice and, although other Smad4 cKO mice also have osteoporosis, these mice later develop increased bone mass[61], while targeted deletion of Smad3 in mice also results in osteopenia due to reduced bone formation, associated with increased osteocyte number[62].

Previous studies have linked TRAF6 to TGFβ-induced activation of p38/MAPK signaling, for example, in prostatic cancer cell migration[36]. Following treatment with TGFβ, TRAF6 induced Lys63-linked polyubiquitination of the p85α subunit of PI3K and Lys63-linked polyubiquitination of TGFβ-activated kinase-1, independent of TGFβRI and II kinase activity. In addition, TGFβ1 enhances and is indispensable for RANKL-induced osteoclastogenesis in vitro through binding of Smad3 to a TRAF6-TAB1-TAK1 complex in OCPs[63], also independent of TGFβR kinase activity.

TRAF3 has important regulatory roles in immune and other cells types that differ from those of other TRAFs, which generally promote NF-κB activation[64]. For example, TRAF3 restrains NF-κB non-canonical activation in OCPs[12] and T and B cells[65]. It also inhibits B cell survival[65], while some human B cell lymphomas[66] and 4–12% of multiple myeloma cases[67] have

inactivating TRAF3 mutations, indicating a tumor suppressor role for TRAF3 in B cells. In addition, mice with TRAF3 deleted in myeloid cells develop early onset osteoporosis[14], inflammatory diseases, infections, and tumors[68], indicating that TRAF3 is an inflammation and tumor suppressor in myeloid cells. Thus, therapeutic approaches to maintain TRAF3 levels in immune cells during aging could prevent or reduce the incidence of several common diseases. Of note, chloroquine prevents ovariectomy-induced osteoporosis in mice[14] and TRAF3 degradation was also prevented by an IAP inhibitor (Fig. 3k). We are currently investigating if prevention of TRAF3 degradation by long-term treatment of aging mice with CQ or IAP inhibitors can prevent age-related osteoporosis.

## Methods

**Animals**. Mice with TRAF3 conditionally knocked out (TRAF3 cKO) in osteoblast lineage cells were generated by crossing *Traf3*[f/f] mice (B6 background)[14,21] with *Prx1*[cre] mice (Jackson Lab #005584). Male and female TRAF3 cKO (*Traf3*[f/f]*Prx1*[cre]) mice and their littermates (*Traf3*[f/f]) were sacrificed at 7 days, 3, 9, 12, and 15 months of age. WT C57BL/6 (B6) mice were from the National Cancer Institute (Frederick, MD, USA). No randomization was done to select animals for study. All animal procedures were conducted in compliance with all applicable ethical regulations using procedures approved by the University of Rochester Committee for Animal Resources.

**Reagents**. The following Abs were purchased from Santa Cruz Biotechnology Inc.: TRAF3 (clone M20, #sc-947), Ubiquitin (clone P4D1, #sc-8017), RelA (clone C20, #sc-372), RelB (clone C19, #sc-226), TGFβ1 (clone V, #sc-146), RANKL (clone FL-317, #sc-9073), HDAC2 (clone H54, #sc-7899), GAPDH (clone 6C5, #sc-32233). These primary Abs were used at following concentrations: Ubiquitin (1:200) and all others (1:500). The following Abs were from Cell Signaling Technology Inc.: β-catenin (#9562), Ser675 phospho-β-catenin (clone D2F1, #4176), GSK3β (clone 27C10, #9315S), Ser9 phospho-GSK3β (clone 5B3, #9323), Smad2 (clone D43B4, #5339), phospho-Smad2 (Ser465/467) (clone 138D4, #3108), Smad3 (clone C67H9, #9523), and phospho-Smad3 (Ser423/425) (clone C25A9, #9520). These primary Abs were used at a concentration of 1:1000. β-actin (clone AC40, #A5441) Ab was purchased from Sigma-Aldrich and used at a concentration of 1:5000. LAMP2 (clone GL2A7, #ab13524) and Tyr216 phospho-GSK3β (#ab75745) Abs from Abcam and OPG Ab (#BAF459) from R&D Systems were used at a concentration of 1:1000. For flow cytometry, APC-conjugated anti-CD45 (clone 104, #17-0454-82), PE-Cy7-conjugated anti-Sca1 (clone D7, #25-5981-82) and Biotin-conjugated anti-RANKL (clone IK22/5, #13-5952–82) Abs were purchased from eBioscience. FITC-BrdU flow kit (#559619) and PE-Texas Red-conjugated streptavidin (#551487) Abs were purchased from BD Biosciences. Primary and secondary Abs for Flow were used at a concentration of 1:100. Recombinant murine TGFβ1 (#7666-MB), TNFα (#410-MT), and BMP2 (#355-BM) were from R&D Systems, and human PTH 1-34 (#3011) from Tocris Bioscience. MG132 (#M8699) and chloroquine (#C6628) were purchased from Sigma-Aldrich. SB-216763 (#HY-12012) was purchased from MedChem Express. ELISA kits for osteocalcin (#LS-F22474) were from LifeSpan BioScience, Inc., for TRACP5b (#MBS763504) from MyBioSource, Inc., and for TGFβ1 (#DY1679-05 for mouse; #DB100B for human) from R&D Systems.

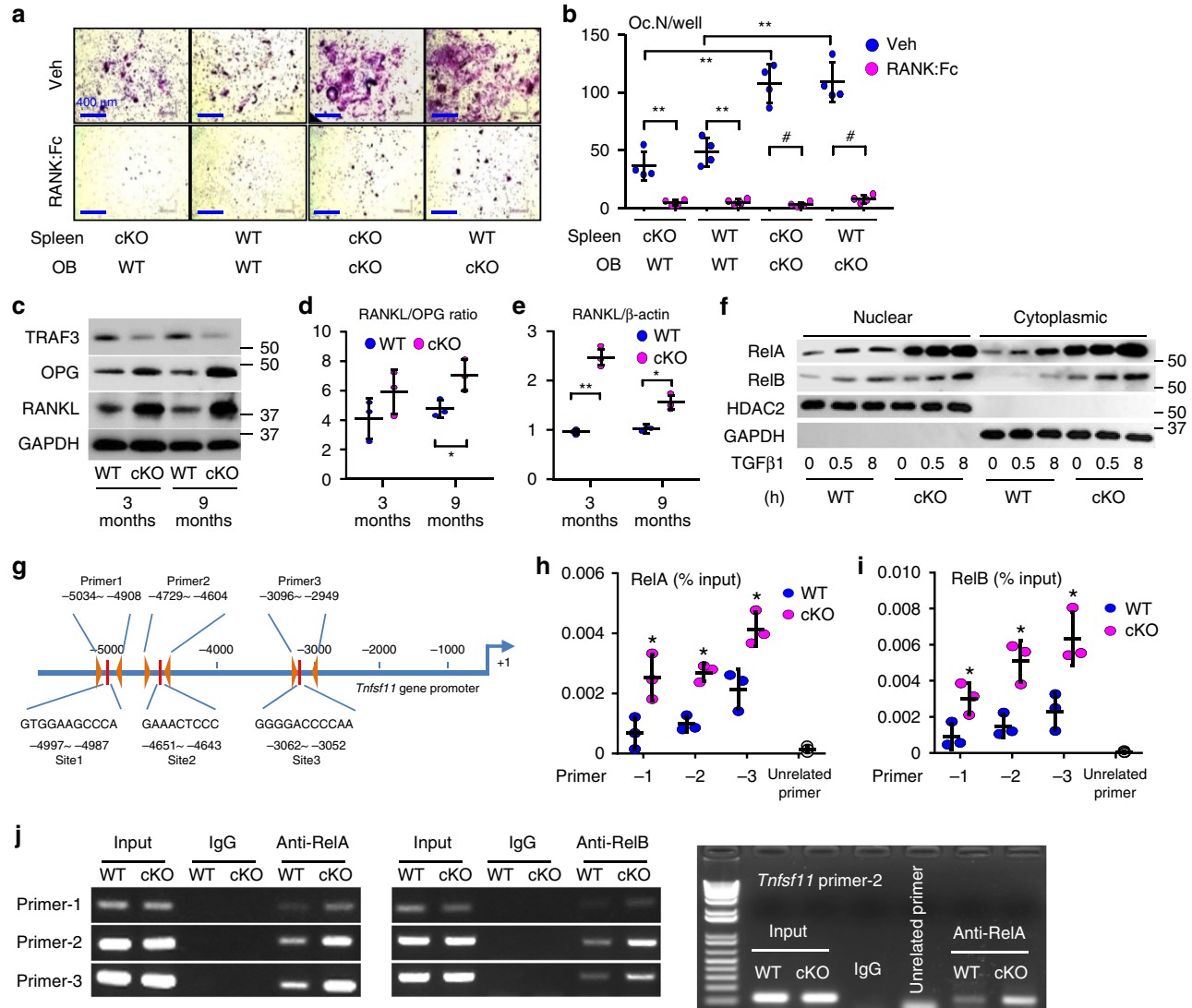

**Fig. 5** RANKL expression is increased in TRAF3 cKO osteoblastic cells. **a** Calvarial pre-OBs and spleen cells isolated from 7-day-old WT or cKO pups, co-cultured with $10^{-8}$ M 1,25(OH)$_2$Vitamin D$_3$ −/+ RANK:Fc (1 μg/ml) for 7 days and TRAP-stained. **b** Osteoclast numbers per well (Oc.N/well) in (**a**) were counted. Mean ± SD ($n = 4$ biologically independent samples; $^\#p < 0.05$, $^{**}p < 0.01$). **c** WB of TRAF3, OPG, RANKL, and GAPDH in protein lysates from tibial metaphyseal bone from 3- and 9-m-old WT and cKO mice. **d** Densitometry analysis of RANKL/OPG protein ratio in (**c**). Mean ± SD ($n = 3$ biologically independent samples; $^*p < 0.05$). **e** RANKL mRNA expression in BdMPCs from 3- and 9-m-old WT and cKO mice. Mean ± SD ($n = 3$ biologically independent samples; $^*p < 0.05$, $^{**}p < 0.01$). **f** BdMPCs from 3-m-old WT and cKO mice treated with TGFβ1 (1 ng/ml) for 0, 0.5, and 8 h. WB of RelA, RelB, HDAC2, and GAPDH in nuclei and cytoplasm. **g** Scheme for mouse RANKL promoter analysis showing putative κB binding sites and primer design to test binding sites. **h**, **i** Sheared chromatin from WT and cKO BdMPCs was used to perform DNA IP using **h** RelA, **i** RelB Abs or IgG control. Real-time PCR performed using designed primers that contain the putative κB binding sites 1, 2, 3, or an unrelated site, normalized to the input. Mean ± SD; $n = 3$ biologically independent samples; $^*p < 0.05$. **j** Sheared chromatin from WT (W) and cKO (K) BdMPCs used to perform DNA IP using RelA, RelB, or IgG control Abs. PCR performed using designed primers that contain the putative κB binding sites 1, 2, 3, and an un-related site. All analyses done using one-way ANOVA with Tukey's post-hoc test. All the in vitro experiments repeated twice with similar results

**Micro-CT and bone histomorphometric analysis**. Following our standard protocol for in vivo assessment of bone formation[12,29], mice were given injections of calcein (10 mg/kg) 5 and 1 days before sacrifice. Right tibiae and T12–L2 vertebrae were fixed in 10% neutral buffered formalin for 2 days. Micro-CT scanning was performed on these specimens using a vivaCT 40 instrument (Scanco Medical) with a resolution of 10.5 μm in a 1 mm section of trabecular bone beneath the growth plates of tibiae and in the entire trabecular bone in the first lumbar vertebrae.

These bone samples were then processed sequentially through 90% ethanol for 2 h (changed every hour), 100% ethanol for 2 h (changed every 1 h), and LR white hydrophilic medium (Polyscience; #17411-500) for 30–36 h (changed every 10–12 h), and then heated at 60 °C overnight to cure the plastic. 4 μm thick unstained sections were cut using a Shandon Finesse ME microtome and examined under fluorescence microscopy for calcein double labeling analysis. Left tibiae and

L3–L5 vertebrae were fixed in 10% neutral buffered formalin for 2 days and processed in 70% ethanol for 2 days, decalcified in 10% EDTA for 14 days and embedded in paraffin. Dynamic parameters of bone formation on unstained plastic sections and static parameters of osteoblasts and osteoclasts on 4 μm thick H&E- and TRAP-stained paraffin sections were blindly assessed by an investigator who was not involved in the sample collection and group assignment using an OsteoMeasure Image Analysis System (Osteometrics, Decatur, GA)[12,29].

**In vitro osteoblast differentiation assay**. To assess osteoblast differentiation capacity, $1 \times 10^6$ BM cells from WT and cKO mice were seeded in 12-well-plates with α-MEM containing 15% FBS for 5 days. This was followed by treatment with osteoblast differentiation medium (50 μg/ml ascorbic acid and 10 mM β-glycerophosphate[29] in α-MEM) for 7 days, with a medium change on day 4. After 7 days, the cells were fixed in 10% neutral buffered formalin followed by ALP

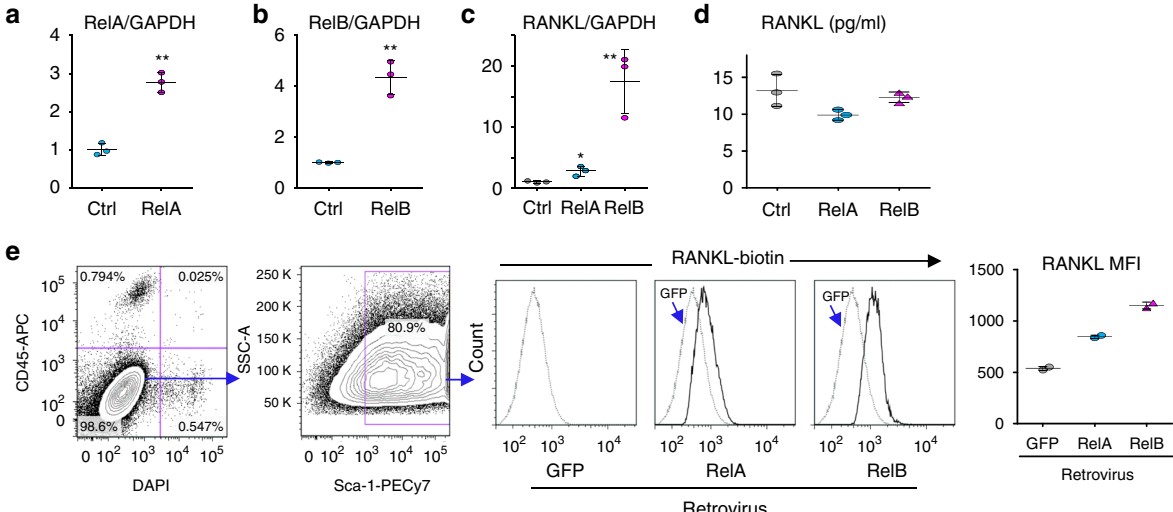

**Fig. 6** RelA and RelB increase RANKL expression by MPCs. Calvarial pre-osteoblasts from 7-day-old WT mice were infected with GFP control, RelA, or RelB retroviruses for 48 h. **a** RelA and **b** RelB mRNA expression tested by real-time PCR to confirm successful over-expression. Mean ± SD ($n = 3$ biologically independent samples; $**p < 0.01$; unpaired Student's $t$ test). **c** RANKL mRNA expression tested by real-time PCR. Mean ± SD ($n = 3$; $*p < 0.05$, $**p < 0.01$; one-way ANOVA with Tukey's post-hoc test). **d** Culture media collected from culture wells and RANKL protein levels measured by ELISA. Mean ± SD ($n = 3$ biologically independent samples; no significant difference; one-way ANOVA with Tukey's post-hoc test). **e** Membrane-bound RANKL levels in 50,000 CD45-Sca-1+ MPCs tested by flow cytometry and expressed as mean fluorescence intensity (MFI). Average of 2 biologically independent samples from two individual experiments. All the in vitro experiments were repeated twice with similar results

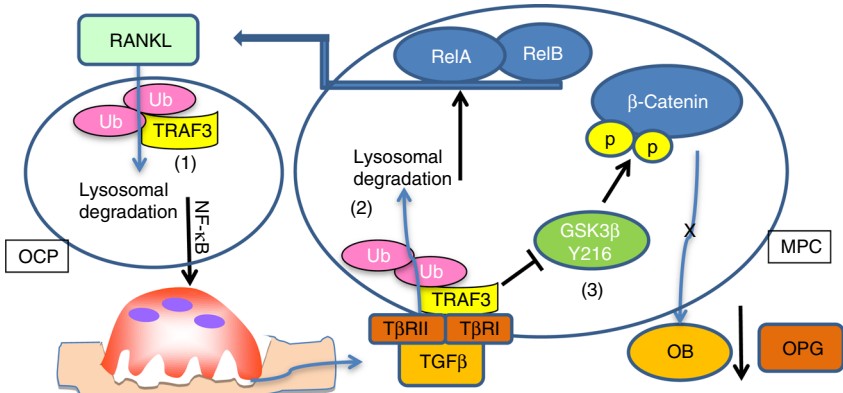

**Fig. 7** Model of TRAF3 degradation during aging leading to bone loss. (**1**) During aging, increased RANKL expressed by MPCs induces TRAF3 ubiquitination and subsequent lysosomal degradation in osteoclast precursors (OCP) to stimulate osteoclast formation and bone resorption through NF-κB[12,14]. As a result, increased amounts of TGFβ are released from bone matrix and activated in the acid environment in resorption lacunae. (**2**) Activated TGFβ induces TRAF3 ubiquitination and subsequent lysosomal degradation in MPCs. As a result, both RelA and RelB are activated to promote RANKL production, further enhancing bone resorption. In addition, (**3**) in MPCs, TRAF3 binds to the TGFβR and negatively regulates GSK-3β activity to prevent β-catenin degradation, allowing β-catenin accumulation and nuclear translocation to maintain osteoblast differentiation and secretion of OPG[52], which limits osteoclast formation. During aging, TGFβ1 degrades TRAF3 and phosphorylates Tyr216 to activate GSK-3β, resulting in degradation of β-catenin to inhibit osteoblast differentiation and OPG secretion to promote osteoclast formation along with increased RANKL expression and bone loss. In young WT mice, TGFβ levels in bone, BM, and serum are not increased and TRAF3 is present to limit these processes

staining to measure the ALP+ cell area (Ar.). Some wells were counter-stained with eosin to measure the total cell Ar. and the ratio of ALP⁺/total cell Ar.

BdMPCs, used to test osteoblast differentiation potential of WT and cKO mice, were generated from tibiae and femora following our reported procedures[29]. Briefly, after BM was completely flushed out with α-MEM, the tibial and femoral bones were cut into small pieces with scissors and cultured with α-MEM containing 15% FBS for 4 days. The bone pieces from each mouse were transferred to a new dish and cultured with α-MEM containing 10% FBS for 4–5 days. Cells that grew out from the bone pieces were digested gently and passaged to new dishes, leaving behind cells that were tightly attached to the dishes and these were discarded. When the passaged cells reached 90% confluence after 2–3 days, they were passaged again and split 1:2. This procedure was repeated once to produce highly pure (>95%) BdMPCs.

To assess osteoblast precursor proliferation, BdMPCs were sequentially cultured from small to larger wells. Initially, $5 \times 10^3$ BdMPCs from 3- and 9-m-old WT and

cKO mice were cultured in each well of a 24-well-plate for 4 days, and cell numbers were counted after digestion with 0.2% of trypsin. The digested cells were passaged to two 12-well-plates. On day 6, the cells were digested and seeded into four 6-well plates. Proliferation was assessed by counting cell numbers on days 1, 2, 4, 6, and 8.

To assess cell cycle status, $5 \times 10^5$ BdMPCs from 3-m-old WT and cKO mice were cultured in 100 mm dishes with α-MEM containing 10% FBS. When they reached ~60% confluence they were starved for 24 h, followed by incubation with BrdU (10 μM) for an additional 8 h. Cell cycle status was analyzed by flow cytometry and the % of cells in S-phase was calculated.

**Osteoblast induction of osteoclastogenesis.** Calvarial pre-osteoblasts and spleen cells from 7-day-old mice were co-cultured to induce osteoclast formation following our published procedures[44,69]. Briefly, cleaned calvarial bones were digested with 0.1% collagenase and 0.2% protease in PBS for 10 min × 6 times and the cells

were collected from digestions 2 to 6; $5 \times 10^3$ cells calvarial cells were seeded in 96-well-plates. Spleens were ground up in a 40 μm cell strainer to make a single cell suspension, which was incubated with red blood cell lysis buffer for 10 min. $5 \times 10^4$ spleen cells were added to the wells containing the calvarial cells and cultured in the presence of $10^{-8}$ M 1,25(OH)$_2$VitD$_3$ for 7 days when large multinucleated cells are observed under inverted microscopy. TRAP staining was performed to evaluate osteoclast formation.

**Over-expression of TRAF3 and GSK-3β.** $5 \times 10^5$ calvarial pre-osteoblasts from newborn WT mice were cultured overnight at 37 °C with 5% CO$_2$ in a 100-mm dish, followed by treatment with 25% volume of pMX-GFP or pMX-TRAF3 retroviral supernatants generated by Plat-E packaging cells and 2 μg/ml polybrene[15]. 48 h later, the retroviral infection medium was changed with fresh cell culture medium (α-MEM containing 10% FBS) plus with PBS or TGFβ1 for 8 or 48 h. The culture medium was discarded and the cells were gently washed twice with PBS. To detect protein phosphorylation, 400 μl protein lysis buffer (RAPI lysis buffer containing protease inhibitor cocktail; (Roche; #11697498001), 1 mM phenylmethylsulfonyl fluoride (PMSF), 50 mM sodium fluoride, 0.2 mM sodium orthovanadate, and 10 μg/ml pepstatin) was added to each 100-mm dish and the cell protein lysate was collected.

Similarly, cells seeded on chamber slides were fixed following retroviral infection and TGFβ1 treatment by 4% PFA at 37 °C for 5 min, and β-actin (cytoskeletal) and β-catenin were detected by immunofluorescence staining[15,70]. The human GSK-3β gene (NM_001146156.1) was cloned to the VB170926-1077hue lentiviral expression vector (Cyagen) with HA-tag. The DNA sequences of TCC for Ser9 and TAT for Tyr216 were point-mutated to GGG (Gly) and GTT (Val), respectively, using a GeneArt Site-Directed Mutagenesis Plus Kit (Life Technologies Corp., #A14604). The mutation was confirmed by DNA sequencing. The lentiviruses were packaged in HEK 293T cells (ATCC, #CRL-3216) using a Lentiviral Packaging Kit (OriGene, #TR30037). As with pMX-TRAF3/GFP retroviral infection, the viral supernatants (25% v/v) were added to $5 \times 10^5$ cultured MPCs with 2 μg/ml polybrene overnight to over-express GFP control and WT or mutated GSK-3β followed by treatment with PBS or TGFβ1 for 48 h. Protein levels of WT, Ser9-, or Tyr216-phospho-GSK-3β, and β-catenin in these cells were tested by WB.

**Collection of human bone samples.** We followed a protocol with informed consent from all patients or their guardians that was approved by the Research Subjects Review Board of the University of Rochester Medical Center. These human studies were performed in adherence to the relevant ethical regulations (Declaration of Helsinki). We collected samples of vertebral bone that were removed from pediatric and adult patients undergoing elective surgery to correct spinal scoliosis and degenerative conditions, including cervical spondylosis, lumbar spinal stenosis, and disc herniation. We used a Rongeur to remove portions of the spinous processes in posterior cervical, thoracic, and lumbar spine procedures. In anterior cervical spine procedures, a Kerrison Rongeur was used to remove bony portions of the anterior overhang of the cervical vertebrae. These bone samples would typically have been discarded as part of the surgical procedure. The study enrolled 55 subjects, including 28 females and 27 males, ranging from 8- to 87-year-old, in which 26 subjects were 8–18-year-old children (10 males, 16 females) and the remaining 29 were middle aged to elderly from 53 to 87 years (18 males, 11 females). Subjects with tumors, active systemic, immunologic, inflammatory, or metabolic disorders that might affect bone remodeling were excluded. Since levels of TRAF3 in human bone samples had not been assessed previously, we were unable to use a power analysis to determine the number of samples that would be required. We estimated that we would require a minimum of 20 samples from children and from adults to detect statistically significant differences in levels between them.

**Western blot analysis.** Human vertebral bone specimens and mouse long bones and vertebrae were ground in liquid nitrogen with a mortar and then lysed with T-Per lysis buffer (Thermo Scientific) containing a protease inhibitor cocktail (Roche, #11697498001). MPCs infected with retrovirus or treated with different reagents were lysed in RIPA lysis buffer (Millipore, #20-188) containing a protease inhibitor cocktail. Protein lysates were incubated on a shaker at 4 °C for 30 min and collected after centrifugation ($16.2 \times 10^3 g$ for 15 min). 10–20 μg of protein lysates were loaded in 10% SDS-PAGE gels and transferred onto polyvinylidene difluoride membranes. Membranes were incubated with the primary Ab overnight followed by incubation with horseradish peroxidase-linked secondary Ab (Bio-Rad) for 2 h. The membranes were exposed to ECL substrate, and signals were detected using a Bio-Rad imaging system. The densitometry analysis was performed using Image Lab 5.1 software by Bio-Rad.

**In vitro ubiquitination assays.** $5 \times 10^5$ BdMPCs were initially seeded in 100-mm dishes and pretreated with chloroquine (100 μM) or MG132 (20 μM) for 4 h before treatment with vehicle or TGFβ1 plus chloroquine or MG132 for 8 h. Cells were lysed in RIPA lysis buffer containing 20 mM HEPES, 250 mM NaCl, 20 mM Tris–HCl, 0.5% NP-40, 2 mM EDTA, 2 μg/ml leupeptin, 2 μg/ml aprotinin, 1 mM DTT, 1 mM PMSF, 1 mM N-ethylmaleimide (Sigma-Aldrich, #E3876) and 1 μg/ml

ubiquitin aldehyde (Enzo Life Sciences, #BML-UW8450-0050) to limit deubiquitination. 500 μg whole cell protein lysates were incubated with anti-TRAF3 Ab, and precipitated proteins were subjected to WB analysis using anti-Ub Ab.

**Enzyme-linked immunosorbent assay (ELISA).** Serum levels of osteocalcin and TRACP5b were tested by ELISA according to the manufacturer's instructions[12,15,29]. Levels of TGFβ1 in mouse serum were measured using a mouse TGFβ1 DuoSet ELISA kit (#DY1679) and in human vertebral specimens using a human TGFβ1 Quantikine ELISA Kit (#DB100B) from R&D Systems according to the manufacturer's guidelines. Briefly, total TGFβ1 levels were measured in 40 μl of serum or 10 μg of vertebral protein lysates by mixing these with 20 μl 1 N HCl, followed by neutralization with 20 μl 1.2 N NaOH/0.5 M HEPES, according to the manufacturer's sample activation procedure; levels of endogenously active TGF-β1 in these samples were measured by skipping this sample activation procedure. These samples were then diluted 1:20 in assay diluent for measurement of total TGFβ1, or they were diluted 1:4 in assay diluent for measurement of active TGFβ1. ELISA plates were analyzed by investigators blinded to sample identity by reading absorbance using a microplate reader set to 450 nm with wavelength correction set to 540 nm.

**Immunofluorescence staining.** Paraffin-embedded sections (4 μm thick) of decalcified L2 vertebrae from 2- and 18-m-old C57BL6/J mice were double-immunostained using Abs to TRAF3 and osteocalcin (Ocn), and covered with Vectashield mounting medium with DAPI (Vector Laboratories, #H-1200,) to visualize nuclei. The numbers of TRAF3/Ocn double-positive cells on trabecular bone surfaces were expressed per mm bone surface and the numbers of TRAF3[+] cells in BM were expressed per mm$^2$ BM in 5 fields (each ~0.15 mm$^2$) that included most of the bone and marrow in the vertebrae using a ×20 objective lens and a Zeiss fluorescence microscope.

BdMPCs from 3-month-old WT mice were cultured on chamber slides (Lab-Tek, #C7182) with 4000 cells per chamber and maintained overnight in α-MEM with 10% FBS. They then were treated with chloroquine (CQ; 50 μM) for 2 h followed by treatment with vehicle or TGFβ1 (1 ng/ml) plus CQ for 8 h, and expression of TRAF3 and LAMP2 was assessed in a minimum of 50 cells in 5 representative fields at ×20 magnification in each of 4 chamber slides in each group.

Calvarial pre-OBs from 7-day-old WT and cKO mice were cultured overnight in α-MEM with 10% FBS on chamber slides (4000 cells per chamber) and then transfected with pMX-GFP or pMX-TRAF3 retroviruses for 48 h followed by treatment of vehicle or TGFβ1 (1 ng/ml) for 48 h. The expression of β-catenin and β-actin was assessed using IF. The area of β-catenin positive staining in nuclei was measured in a minimum of 50 cells in 5 representative fields using a ×20 objective lens in each of 4 chamber slides in each group.

The above cell cultures were terminated by fixation with 4% paraformaldehyde for 5 min at 4 °C, and cells were permeabilized with 0.1% Triton X-100 and blocked with 5% goat serum for 1 h at room temperature before incubation with primary Abs in a humidity chamber at 4 °C overnight. On the 2nd day, after 3 washes with PBS containing 0.05% Tween 20, Alexa Fluor 488- and 568-conjugated secondary Abs were added and incubated for 1 h at room temperature. A minimum of 50 stained BdMPCs and calvarial cells were imaged in 5 representative fields in each of 4 chamber slides in each group using a ×20 objective lens and a Zeiss fluorescence microscope. These fields were photographed and the areas of co-localization of TRAF3 and LAMP2 in the cytoplasm of BdMPCs and of β-catenin positive staining in nuclei of calvaria-derived cells were measured in the digital images using an ImageJ image analysis system (NIH).

**ChIP assay.** Transcription factor binding sites within −8 kb before the murine RANKL coding start site were searched using TFSEARCH software to identify putative κB binding sites. ChIP assays were performed to test for binding of RelA and RelB to each of the three κB binding sites, following our published procedure[29]. Briefly, the sheared chromatin from WT and TRAF3 cKO BdMPCs that had been fixed with 1% formaldehyde was immunoprecipitated with Abs to RelA or RelB, or rabbit IgG as a negative control. The precipitated DNA was used as a template for PCR using primers specifically designed to amplify a segment of 120–150 bp containing the putative κB binding sites. The sequences of the primers are: site 1, forward 5′-TGCTGCAATCCTTTAACACA-3′ and reverse 5′-CCCCT TGGGAGATATCAGA-3; site 2, forward 5′-CTGCCATGTTGTTCAGCCTA-3′ and reverse 5′-AGGAGGAAAAACAGGGTCCTT-3; and site 3, forward 5′-GTGG TTGGAGTCTACCATGC-3′ and reverse 5′-CCCCATGAGTGGATAGATGC-3. In addition, a pair of unrelated primers, forward 5′-AAGAAGCCTAGAGTCCC TGG-3′ and reverse 5′-CCTGCGACAGCGGAGAAAAG-3, was designed in the DNA region that is outside the 2.5 kb of the κB binding sites.

**Quantitative real-time PCR.** 1 μg of total RNA extracted from MPCs and cortical bones was reversely transcribed to cDNA in a 20 μl reaction using an iSCRIPT cDNA Synthesis kit (Bio-Rad). The expression levels of *Tnfsf11* (encodes RANKL) and genes that are involved in perilacunar/canalicular remolding, including *Acp5, Mmp2, Mmp13, Mmp14,* and *Ctsk*, were measured using an iCycler real-time PCR

machine (Bio-Rad) with iQ SYBR SuperMix (Bio-Rad) according to the manufacturer's instruction.

Primer sequences are as follows: *Tnfsf11*, forward, 5′-CAGAAGGAACTGCAA CACAT-3′, and reverse, 5′-CAGAGTGACTTTATGGGAACC-3′; tartrate-resistant acid phosphatase type 5 (*acp5*), forward, 5′-TCCTGGCTCAAAAA GCAGTT-3′, and reverse, 5′-ACATAGCCCACACCGTTCTC-3′; cathepsin K (*ctsk*), forward, 5′-CAGCTTCCCCAAGATGTGAT-3′, and reverse, 5′-GAAGC ACCAACGAGAGGAGA-3′; *Mmp2*, forward, 5′-AACGGTCGGGAATACAG CAG-3′, and reverse, 5′-GTAAACAAGGCTTCATGGGG-3′; *Mmp13*, forward, 5′-CGGGAATCCTGAAGAAGTCTACA-3′, and reverse, 5′-CTAAGCCAAAGAA AGATTGCATTTC-3′; *Mmp14*, forward, 5′-AGGAGACGGAGGTGATCATCAT TG-3′, and reverse, 5′-GTCCCATGGCGTCTGAAGA-3′; *Gapdh*, forward, 5′-GGTCGGTGTGAACGGATTTG-3′, and reverse, 5′-ATGAGCCCTTCCACAA TG-3′.

The relative abundance (ΔCT) of each gene was calculated by subtracting the GAPDH CT value from the corresponding CT value of specific genes, and ΔΔCT values were obtained by subtracting the ΔCT values of WT samples from cKO samples, and then raised to the power 2 ($2^{-\Delta\Delta CT}$) to yield fold-expression relative to the WT controls.

**Flow cytometry.** Calvarial pre-osteoblasts from 7-day-old WT mice were infected with GFP control, pMX-RelA-GFP or pMX-RelB-GFP retroviruses for 2 days before being harvested for FACS analysis. Briefly, the live cells were stained with anti-CD45-APC, anti-Sca1-PECy7 and anti-RANKL-Biotin Abs in FACS buffer (2% FBS in PBS) at 4 ℃ for 30 min, followed by PE-Texas Red Streptavidin staining at 4 ℃ for 30 min. After the staining was completed, cells were kept in FACS buffer with DAPI and tested immediately. Around 50,000 CD45⁻Sca1⁺ live cells were gated out for further analysis of surface RANKL expression. To measure BrdU⁺ S-phase BdMPCs, a FITC BrdU flow kit (#559619) from BD Bioscience was used for cell staining. Briefly, cells were fixed and permeabilized using BD Cytofix/Cytoperm buffer and permeabilization buffer plus, then digested in 300 μg/ml DNase for 1 h at 37 ℃. Each sample of cells was stained with 1 μl FITC-conjugated BrdU antibody in 50 μl BD Perm/Wash buffer for 20 min at room temperature and after washing once were resuspended in 20 μl 7-AAD. Stained cells were acquired using a flow cytometer (FACS LSR II; BD Biosciences). FlowJo software was used for data analysis.

**Statistics**. All results are given as the mean ± S.D. Variance was similar between groups for most parameters assessed. Comparisons between two groups were analyzed using Student's two-tailed unpaired $t$ test and those among 3 or more groups using one-way analysis of variance followed by Tukey's post-hoc multiple comparisons. $p$ Values < 0.05 were considered statistically significant. Each experiment was repeated at least twice with similar results. The sample size for in vivo experiments is based on an un-paired $t$-test power analysis carried out by our statistician using SigmaStat Statistical Software: 5–8 mice were needed in each group where bone parameters are being assessed to detect significant differences from controls with an alpha error of 5%. The power is 0.98, i.e., there is 98% chance of detecting a specific effect with 95% confidence when alpha = 0.05. No data were excluded from the analyses.

## Data availability

All relevant data are available from the authors upon reasonable request.

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

## Acknowledgements

Research reported in this publication was supported by the National Institute of Arthritis and Musculoskeletal and Skin Diseases of the National Institutes of Health under Award Numbers R01AR43510 (to B.F.B.), P30 AR069655 (to Edward Schwarz), AR063650 (to L.X.) and by the National Institute for Aging under Award Number R01AG049994 (to B.F.B. and Z.Y.), and 1S10RR027340 (to B.F.B.) from the National Institutes of Health. The content is solely the responsibility of the authors and does not necessarily represent the official views of the National Institutes of Health. The TRAF3ff/ff mice were a gift from Dr. Gail Bishop, University of Iowa.

## Author contributions

J.L. performed and analyzed the experiments shown in Figs. 1a–i, 2e–g, 3a–e, g–n, p, 4a–m, 5c–e, g–j, as well as Fig. 6 and wrote the first draft of the paper. A.A. performed and analyzed the experiments shown in Figs. 2a–d, h–m and 5f. Y.X. generated the TRAF cKO mice and carried out initial skeletal analyses. X.Y. performed and analyzed the experiments shown in Fig. 3f, o. J.O.S. and A.M. provided human bone samples. Z.Y. conceived and coordinated the studies, performed and analyzed the experiments shown in Figs. 1a, b, d–f, h, 3e, 4g, 5a, b, and wrote the paper. B.F.B. supervised, conceived and coordinated the studies, and wrote the paper. All authors reviewed the results and approved the final version of the manuscript.

## Additional information

**Competing interests:** The authors declare no competing interests.

