## [Peer Review File · Nature Communications]

Reviewers' Comments:

Reviewer #1:

Remarks to the Author:

This is an interesting report characterizing a new mouse strain with a novel phenotype. Parts of the report present convincing new findings regarding an important role for TRAF3 in regulation of osteoblast differentiation. The final Figures are also straightforward and reasonable. However, some portions (especially Figs. 3 and 4) are problematic, over-interpreted, and in some instances, based upon flawed reasoning. Specific comments are as follows.

This new mouse strain lacking TRAF3 in bone precursors has a striking phenotype, characterized in Fig. 1. The overall impression given is that absence of TRAF3 accelerates normal aging-related changes in bone, such that the greatest impact is seen in 'late middle-aged' 9-month mice, while by 15 months similar changes are being found in WT mice. The only concern I have with Fig. 1 is that the differences shown by histology in 1f and 1h are difficult to appreciate – were the quantitative data shown in adjoining panels scored by an assessor who was blinded as to the identity of each sample? That would be important.

Fig. 2 introduces the concept that TGFb1 plays an important role in the bone phenotype of this conditional TRAF3-deficient mouse. Here too the data are reasonable and make a good case for this interpretation.

In Fig. 3 data and their interpretation become more complex. It is a plus that the authors had access to human bone samples as a complementary approach. However, did the samples shown in the 3b Western blot contain hematopoietic cells? TRAF3 will be expressed in such cells, and it may also be reduced in this compartment with aging (see Molony et al, Science Signaling 2017); additionally, aged individuals in general have fewer hematopoietic cells. 3f and 3g are intriguing, but over-interpreted. Unless mass spectrometry is performed (to identify ubiquitinated peptides), it is impossible to know if the smears of protein recognized by anti-Ub Ab in IPs represent the precipitated protein itself, or one or more associated proteins. Also, the AT406 inhibitor is presumed to act by blocking cIAP action, so why is LESS TRAF3 seen in the lanes with this drug in the lower panel of 3g? An important omission is a blot for the TGFbR1 itself – is this receptor ubiquitinated? The authors discuss these data as though they assume TRAF3 associates with the receptor – why not investigate directly if this happens? The data in 3i are so dim as to be uninformative.

Fig. 4 presents the weakest data of the paper. It starts reasonably, showing in 4a that TRAF3 and b-catenin decrease in parallel as mice age, and that TGFb1 can induce both to decrease. But 4c is uninterpretable, especially as the levels of TRAF3 in the various transduced cells are never shown. The data quantified in the right panel are hard to relate to what is shown in the left panel. The largest problem is the rationale and interpretation of 4d and 4e. It is well-documented in publications that GSK3 phosphorylates b-catenin in a manner that promotes its degradation, while Akt PROMOTES b-catenin function as a transcription factor in two ways. One is that Akt can phosphorylate GSK3 on an inhibitory residue. Akt can also phosphorylate b-catenin itself, but on a different residue from that used by GSK3, so Akt-mediated b-catenin phosphorylation is activating, and does not promote degradation. Thus, the authors' model in which TRAF3 promotes b-catenin activity by inhibiting Akt is inconsistent with what is already known. 4d shows enhanced amounts of total AKT following 8h of TGFb1 treatment, which should reduce TRAF3. If less TRAF3 leads to more AKT, however, why is no AKT seen in cKO samples, regardless of TGFb1 treatment? With no blot for TRAF3 or AKT, 4e is uninterpretable, as is 4f. Thus, their model just does not fit with either their own, or considerable published data. The SB inhibitor used in 4g is actually a GSK3 inhibitor, a kinase never mentioned in their discussion of these data.

Fig. 5 returns to a reasonable hypothesis, the TRAF3 deficiency promotes RANKL production via NF-kB2 activation. As the connection between TRAF3 and NF-kB2 is well-documented for other cell

types, this is not unexpected, but the authors do a good job of establishing this, in both Figs. 5 and 6. It should be noted that the ChIP analysis-related PCR performed in 5e is not actually quantitative, so bands shown there should not be over-interpreted.

Overall, the authors have convinced me that TRAF3 is important for osteoblast differentiation, and there is an interesting connection between TRAF3 and TGF β 1 that is worth additional exploration. I also readily believe, as it is certainly consistent with known TRAF3 functions, that TRAF3 deficiency promotes RANKL production via the NF- κ B2 transcriptional pathway. But the rest of what is presented here detracts from these findings, rather than enhancing them.

Reviewer #2:

Remarks to the Author:

Li et al. showed that conditional deletion of TRAF3 in mesenchymal progenitor cells in Prx1-Cre Traf3 cKO mice resulted in osteoporosis from 9 month of age due to decreased bone formation and increased bone resorption. They further showed that TGF- β induced TRAF3 degradation in mesenchymal progenitor cells and inhibited osteoblast differentiation by inducing β -catenin degradation through Akt-mediated pathway in vitro. Importantly, they also showed that TRAF3 protein level in bone was decreased in aged humans. Overall, the observation of TRAF3 function in osteoblast precursor cells is novel, but the authors could not provide compelling experimental data to support the authors' hypothesis in the manuscript.

Major concerns

- 1) The authors proposed that the increase of serum TGF- β level in aged cKO mice led to decreased osteoblast differentiation and increased osteoclast differentiation. However, most of the previous studies indicated that serum level of TGF- β decreased with age in human (Kamio K et al., *Geriatr Gerontol Int*, 2013; Redondo S et al., *J Cardiothorac Surg*, 2014; Zhang N et al., *Clin Chim Acta*, 2009). To substantiate the hypothesis, the authors should investigate whether the phenotype of cKO mice can be ameliorated by the inhibition of TGF- β signaling (by the administration of the anti-TGF- β antibody or an inhibitor for TGF- β receptor, such as SD208 and SB525334). Preferably, the authors should analyze TGF- β level in bone from healthy older subjects compared with younger ones.
- 2) It was previously reported that the increase of serum TGF- β level can induce skeletal muscle weakness by decreasing Ca²⁺-induced muscle force production (Waning DL et al., *Nat Med*, 2015). Muscle forces has been considered to provide mechanical stimuli on bone and bone can adapt its structure in response to muscle-derived stimuli. Therefore, the authors should investigate the effect of excess TGF- β level in cKO mice on muscle weakness.
- 3) A previous report demonstrated that Smad4 ablation led to a decreased osteoblast differentiation, along with an increased rate of cell proliferation in vivo (Salazar VS et al., *J Cell Sci*, 2013). Therefore, Smad4-deficiency in osteoblasts seems to recapitulate the phenotype of TRAF3 cKO cells. In addition, another report showed that RelB interacts with Smad2/4 to repress transcriptional activity of Smads and this was inhibited by TRAF3 (Newman AC, et al., *Nat Commun*, 2017). Nevertheless, the authors did not analyzed the involvement of TGF- β -induced Smad pathway at all. The authors should investigate the contribution of Smad-dependent signaling on TRAF3 pathway in osteoblast proliferation and differentiation.
- 4) Although the authors showed that TGF- β inhibited osteoblast differentiation via the activation of Akt, the previous study, by using Akt-deficient mice, clearly demonstrated that Akt enhanced osteoblast differentiation by activating Runx2 transcriptional activity and inhibited apoptosis (Kawamura N et al., *PLOS ONE*, 2007). The authors need to explain these discrepancies.

5) It was reported that osteocyte-intrinsic TGF-beta signaling induces genes involved in perilacunar/canalicular remodeling (Dole NS, et al., Cell Rep, 2018). To examine the involvement of TRAF3 in TGF-beta signaling in osteocytes, the authors should analyze the expression of genes involved in perilacunar/canalicular remodeling, along with RANKL/OPG expression in osteocytes, since TRAF3 is also deficient in osteocytes of Prx1-Cre Traf3 cKO mice.

6) In Fig. 3g, AT406, an antagonist of IAP, completely abrogated the binding of TRAF3 to TGF-beta receptor 1. In contrast, AT406 treatment only partially restores osteoblast differentiation as shown in Fig. 3h. This indicated that IAP-independent pathway may be involved in TGF-beta induced inhibition of osteoblast differentiation. The authors should discuss TRAF3-independent pathway during TGF-induced inhibition of osteoblast differentiation.

Minor points

1) Generally speaking, osteoclast number and activity in mice is reduced with age due to aging-associated low turnover of bone remodeling. The authors need to explain why there were no differences in osteoclast number, surface and serum TRAcP5b level between young and aged WT mice in Fig. 1h.

2) The authors described that TRAF3 overexpression prevented TGF-beta-induced beta-catenin phosphorylation. However, in Fig. 4f, the data clearly showed that TGF-beta markedly induced beta-catenin phosphorylation without activation of Akt in TRAF3 ectopically expressed cells. This indicate that TGF-beta-induced Akt-independent pathway may have an major influence on the regulation of beta-catenin. The authors need to explain these points.

3) In Fig. 5d and 5e, RelA and RelB localized nucleus and recruited to RANKL promoter without TGF-beta stimulation even in WT cells. The authors should discuss how NF-kappaB signaling was activated in mesenchymal precursor cells in steady state.

4) In Fig. 4e, the authors performed IP-western assay by using anti-PI3K antibody. However, there were no descriptions of the antibody against PI3K and immunoprecipitation assay in Method section.

We thank the reviewers for their careful and helpful review of our paper and for their suggestions to improve it and clarify the mechanisms whereby we propose that TGF β inhibits osteoblast formation during aging. Our responses to the concerns are listed below. We have made significant revisions to Figures 3 and 4, as suggested, and to the text associated with these changes in the Results section and Figure legends, and these are highlighted in blue in the text of the revised manuscript. We have also added letters to the most of the sub-panels in most of the figures to better conform to the Nat. Communications format, but have not highlighted most of these or associated minor changes in the text to make it easier for the reviewers to identify the major changes we have made to the manuscript.

Reviewer #1 (Remarks to the Author):

1. This new mouse strain lacking TRAF3 in bone precursors has a striking phenotype, characterized in Fig. 1. The overall impression given is that absence of TRAF3 accelerates normal aging-related changes in bone, such that the greatest impact is seen in ‘late middle-aged’ 9-month mice, while by 15 months similar changes are being found in WT mice. The only concern I have with Fig. 1 is that the differences shown by histology in 1f and 1h are difficult to appreciate – were the quantitative data shown in adjoining panels scored by an assessor who was blinded as to the identity of each sample? That would be important.

The data, including those in Fig. 1f and 1h, were assessed in a blinded fashion by an experienced investigator who was not involved in the sample collection or group assignment. This is now stated in the Material and Methods section, “bone histomorphometric analysis” on page 21. We have also replaced the images with ones that better illustrate the differences in the numbers of OCs and OBs on the trabecular surfaces.

2. Fig. 2 introduces the concept that TGF β 1 plays an important role in the bone phenotype of this conditional TRAF3-deficient mouse. Here too the data are reasonable and make a good case for this interpretation.

We appreciate the positive comment.

3. In Fig. 3 data and their interpretation become more complex. It is a plus that the authors had access to human bone samples as a complementary approach. However, did the samples shown in the 3b Western blot contain hematopoietic cells? TRAF3 will be expressed in such cells, and it may also be reduced in this compartment with aging (see Molony et al, Science Signaling 2017); additionally, aged individuals in general have fewer hematopoietic cells.

We agree that aged individuals in general have fewer hematopoietic cells in bone marrow than children, particularly in long bones, and this could have contributed significantly to the reduction in TRAF3 levels we detected in the vertebral spinous bone samples from the aged individuals. We could not and did not attempt to separate BM from the bone in these samples. However, we had excess tissue from 3 children and 3 adults and embedded these bone samples in paraffin after fixation and decalcification. We H&E-stained the sections and measured the BM area/total area and found no significant differences in the BM areas, suggesting that BM may be retained in these vertebral bones during aging. Since the numbers of these samples are small, we have not included these new data in the revised manuscript.

However, we did assess TRAF3 levels in BM and cortical bone from young and aged WT mice and found that they were lower in both the cortical bone and BM. We have included these new data in a revised Fig. 3a and Supplemental Fig. 2a. We also cut sections of vertebrae from 2- and 19-m-old WT mice to do additional assessment of Ocn- and TRAF3-positive cells because these have more trabecular bone than the sections of femoral metaphyses from 19-m-old mice that we assessed in the original submission. We measured the numbers of TRAF3+, mainly hematopoietic, cells in BM in sections of these vertebrae as well as Ocn/TRAF3 double-positive OBs on bone surfaces and found that these were all significantly reduced in the old mice. We have included these new data in a revised Fig. 3c and d, and have referenced the Molony paper. We have also illustrated how reduced TRAF3 levels in osteoclast precursors in BM promote RANKL-induced osteoclastogenesis in Fig.7.

3. 3f and 3g are intriguing, but over-interpreted. Unless mass spectrometry is performed (to identify ubiquitinated peptides), it is impossible to know if the smears of protein recognized by anti-Ub Ab in IPs represent the precipitated protein itself, or one or more associated proteins.

We agree. Since many publications have already shown that TRAF3 degradation is controlled by ubiquitination and de-ubiquitination in B cells, T cells, and bladder epithelial cells¹⁻⁵, for example, it is most likely that TGF β -induced TRAF3 degradation is also due to its ubiquitination. To attempt to demonstrate ubiquitinated TRAF3 by mass spectrometry, we pulled down TRAF3 from lysates of MPCs treated with TGF β + chloroquine (CQ) with the anti-TRAF3 antibody and submitted the TRAF3-IP complex for mass spectrometric analysis. Mass spectrometry confirmed the presence of TRAF3 protein in our IP samples, but it did not detect ubiquitinated peptides that matched the TRAF3 protein sequence. Our mass spectrometry lab director and we think that this is likely due to low levels of ubiquitinated peptides in the IP samples. We have not included these negative data in the revised manuscript. However, to further investigate this point, in a separate experiment, we pulled down ubiquitinated proteins with the anti-ubiquitin antibody and blotted the IP complex with the anti-TRAF3 antibody. This assay demonstrated increased amounts of ubiquitinated TRAF3 proteins in TGF β + CQ-treated cells. We have included these new data in a revised Fig. 3i.

4. Also, the AT406 inhibitor is presumed to act by blocking cIAP action, so why is LESS TRAF3 seen in the lanes with this drug in the lower panel of 3g? An important omission is a blot for the TGF β R1 itself – is this receptor ubiquitinated? The authors discuss these data as though they assume TRAF3 associates with the receptor – why not investigate directly if this happens?

We agree that the data were not compelling. AT406 exerts its biological effects through degradation of cIAP1 and cIAP2^{6,7}. We added WBs of cIAP2 in a revised Fig. 3j and k. Less TRAF3 is seen in the lanes treated with AT406 in the lower left panel of 3j because TRAF3 binding to TGF β R1 requires cIAP1 and 2. We added a blot in the lower right panel of 3j showing that TGF β R1 is present in the cells and confirmed this following immunoprecipitation with the anti-TGF β R1 Ab. Previous studies have reported that phosphorylation of TGF β R1⁸ is associated with its ubiquitination and we have demonstrated that TRAF3 associates with TGF β R1 in Fig. 3j, left lower panel. We have described the results of these new data in an extensively edited section of the manuscript on pages 9 and 10.

5. The data in 3i are so dim as to be uninformative.

We repeated this experiment and replaced the images in a revised Fig. 3m with new brighter ones and have revised the quantification to now depict the area of TRAF3/Lamp2 double-positive staining in the cells in a revised Fig. 3n.

6. Fig. 4 presents the weakest data of the paper. It starts reasonably, showing in 4a that TRAF3 and b-catenin decrease in parallel as mice age, and that TGFb1 can induce both to decrease. But 4c is uninterpretable, especially as the levels of TRAF3 in the various transduced cells are never shown. The data quantified in the right panel are hard to relate to what is shown in the left panel.

We agree and have added a Western blot showing TRAF3 levels in the WT and cKO calvarial cells infected with pMX-GFP (as control) or pMX-TRAF3 retroviruses in a new Fig. 4e. We have redone the quantification of the β -catenin staining in nuclei and have now expressed this as the mean area of β -catenin staining in nuclei in a revised Fig. 4d. These data now relate to the images in 4c.

7. The largest problem is the rationale and interpretation of 4d and 4e. It is well-documented in publications that GSK3 phosphorylates b-catenin in a manner that promotes its degradation, while Akt PROMOTES b-catenin function as a transcription factor in two ways. One is that Akt can phosphorylate GSK3 on an inhibitory residue. Akt can also phosphorylate b-catenin itself, but on a different residue from that used by GSK3, so Akt-mediated b-catenin phosphorylation is activating, and does not promote degradation. Thus, the authors' model in which TRAF3 promotes b-catenin activity by inhibiting Akt is inconsistent with what is already known. 4d shows enhanced amounts of total AKT following 8h of TGFb1 treatment, which should reduce TRAF3. If less TRAF3 leads to more AKT, however, why is no AKT seen in cKO samples, regardless of TGFb1 treatment? With no blot for TRAF3 or AKT, 4e is uninterpretable, as is 4f. Thus, their model just does not fit with either their own, or considerable published data. The SB inhibitor used in 4g is actually a GSK3 inhibitor, a kinase never mentioned in their discussion of these data.

We thank the reviewer for pointing out these discrepancies/inconsistencies and insights. We agree that TRAF3 regulation of β -catenin via PI3K/AKT is not appropriate and inconsistent with our own and published data.

Our interpretation of some of our early findings was that TRAF3 does indeed regulate β -catenin degradation via GSK-3 β , but our original findings that TRAF3 cKO MPCs did not have increased GSK3 β Ser-9 phosphorylation led us to think that TRAF3/TGF β 1 regulation of β -catenin would bypass GSK3 β , which we now agree was wrong.

Thus, following the reviewer's suggestions, we repeated our experiments, and our new data show that TRAF3 cKO bone has increased GSK3 β Tyr216 phosphorylation, which promotes the activity of GSK3 β . We then mutated Ser9 and Tyr216 in GSK3 β -expressing vectors and infected these constructs into MPCs. Our new findings indicate that over-expression of the Tyr216-mutated GSK3 β increased total β -catenin levels in vehicle-treated cells and prevented TGF β 1-induced β -catenin degradation. Importantly, over-expression of TRAF3 reduced GSK3 β Tyr216 phosphorylation and the association of TGF β R1 with GSK3 β . We have included these new

GSK3 β data in the revised Fig. 4f, g and h, and have deleted the PI3K/AKT story in the revised manuscript.

Based on our new data in Fig. 4, we propose a new molecular mechanism for TRAF3-mediated OB differentiation: in MPCs cells in aged WT mice, TGF β binds to the TGF β R, which phosphorylates GSK3 β on tyrosine 216, leading to β -catenin degradation and inhibition of osteoblast formation. TRAF3 limits this process in young mice. In the absence of TRAF3 or when TRAF3 levels are low, such as during aging, the inhibitory effect of TRAF3 on TGF β -induced β -catenin degradation is reduced, resulting in more GSK3 β Tyr216 phosphorylation, β -catenin degradation, inhibition of osteoblast formation and decreased OPG expression by these cells, which is now illustrated in Fig 7.

8. Fig. 5 returns to a reasonable hypothesis, the TRAF3 deficiency promotes RANKL production via NF-kB2 activation. As the connection between TRAF3 and NF-kB2 is well-documented for other cell types, this is not unexpected, but the authors do a good job of establishing this, in both Figs. 5 and 6. It should be noted that the ChIP analysis-related PCR performed in 5e is not actually quantitative, so bands shown there should not be over-interpreted.

We thank the reviewer for these comments and have added quantitative PCR ChiP data in a revised Fig. 5h and i.

9. Overall, the authors have convinced me that TRAF3 is important for osteoblast differentiation, and there is an interesting connection between TRAF3 and TGF β 1 that is worth additional exploration. I also readily believe, as it is certainly consistent with known TRAF3 functions, that TRAF3 deficiency promotes RANKL production via the NF-kB2 transcriptional pathway. But the rest of what is presented here detracts from these findings, rather than enhancing them.

We thank the reviewer for these supportive comments and really helpful suggestions and feel that addition of the new data showing TGF β 1/TRAF3 regulation of β -catenin via GSK3 β provides a more compelling story with which we hope the reviewer agrees.

Reviewer #2 (Remarks to the Author):

Li et al. showed that conditional deletion of TRAF3 in mesenchymal progenitor cells in Prx1-Cre Traf3 cKO mice resulted in osteoporosis from 9 month of age due to decreased bone formation and increased bone resorption. They further showed that TGF-beta induced TRAF3 degradation in mesenchymal progenitor cells and inhibited osteoblast differentiation by inducing beta-catenin degradation through Akt-mediated pathway in vitro. Importantly, they also showed that TRAF3 protein level in bone was decreased in aged humans. Overall, the observation of TRAF3 function in osteoblast precursor cells is novel, but the authors could not provide compelling experimental data to support the authors' hypothesis in the manuscript.

Major concerns

1) The authors proposed that the increase of serum TGF-beta level in aged cKO mice led to decreased osteoblast differentiation and increased osteoclast differentiation. However, most of

the previous studies indicated that serum level of TGF-beta decreased with age in human (Kamio K et al., Geriatr Gerontol Int, 2013; Redondo S et al., J Cardiothorac Surg, 2014; Zhang N et al., Clin Chim Acta, 2009). To substantiate the hypothesis, the authors should investigate whether the phenotype of cKO mice can be ameliorated by the inhibition of TGF-beta signaling (by the administration of the anti-TGF-beta antibody or an inhibitor for TGF-beta receptor, such as SD208 and SB525334). Preferably, the authors should analyze TGF-beta level in bone from healthy older subjects compared with younger ones.

We agree with the reviewer that previous publications have reported that serum levels of total TGFβ generally decrease during aging in humans, and of course in serum TGFβ1 exists mainly in its latent form, which does not have biological activity until it is activated.

We agree that one way to investigate if increased active TGFβ1 leads to early onset osteoporosis in TRAF3 cKO mice would be to demonstrate that the bone phenotype of the cKO mice is prevented by TGFβ1 inhibition. However, it would take about 12 months in total to do that experiment and get the results after tissue processing and analysis. Instead, as suggested, we measured total and active TGFβ1 levels in the human bone samples from children and adults and have included the results showing that active, but not total, TGFβ1 levels are higher in samples from adults than from children in the revised Figure 3h. We have included a description of these new data on page 9 of the revised manuscript.

2) It was previously reported that the increase of serum TGF-beta level can induce skeletal muscle weakness by decreasing Ca²⁺-induced muscle force production (Waning DL et al., Nat Med, 2015). Muscle forces has been considered to provide mechanical stimuli on bone and bone can adapt its structure in response to muscle-derived stimuli. Therefore, the authors should investigate the effect of excess TGF-beta level in cKO mice on muscle weakness.

We agree that high levels of the active form of TGFβ1 released from bone during extensive tumor-induced osteolysis can travel in the serum and affect muscles, resulting in atrophy, which in turn produces factors, either mechanical or chemical, which could affect the progression of bone loss. Waning et al. used a TGFβ Ab to assess what we assume to be total TGFβ levels in serum. They did not specify if they assessed total and/or active TGFβ, but it is most likely that the more than doubling of TGFβ serum levels they reported was due to release of active TGFβ from the osteolytic lesions in the mice. To attempt to address the reviewer's concern, we looked for possible effects of TGFβ1 on muscle in 12-month-old WT (f/f) and cKO mice. We found that the mass/weight of tibialis anterior, soleus, gastrocnemius, and quadriceps muscles were similar in the mice, and importantly TRAF3 protein levels in these muscles were also similar in the cKO and WT mice, suggesting that prx1-Cre did not affect TRAF3 expression in these muscles. In addition, the cKO and littermates have similar body weights and lengths. However, the muscle fiber diameter was lower and fiber numbers were higher in tibialis anterior muscles from the cKO mice than those from WT littermate mice, but not in other muscles. The numbers of MyHCIIA and MyHCIIB fibers were similar in the cKO and WT mice. We are planning to test muscle strength and function in the cKO mice, but have not yet been able to develop the required expertise. We have chosen not to include these muscle data in this manuscript because they are preliminary and we do not know if the muscle changes are a result of or because of the bone phenotype. We plan to examine in a new grant proposal how muscle

and bone might interact through TRAF. The release of active TGF β 1 from bone is one possible mechanism, but the amounts of TGF β 1 released from bone during aging are likely to be much less than those released from bones of the mice in Waning's study. We hope that the reviewer will agree that retaining our new data for another publication is reasonable.

3) A previous report demonstrated that Smad4 ablation led to a decreased osteoblast differentiation, along with an increased rate of cell proliferation in vivo (Salazar VS et al., J Cell Sci, 2013). Therefore, Smad4-deficiency in osteoblasts seems to recapitulate the phenotype of TRAF3 cKO cells. In addition, another report showed that RelB interacts with Smad2/4 to repress transcriptional activity of Smads and this was inhibited by TRAF3 (Newman AC, et al., Nat Commun, 2017). Nevertheless, the authors did not analyze the involvement of TGF-beta-induced Smad pathway at all. The authors should investigate the contribution of Smad-dependent signaling on TRAF3 pathway in osteoblast proliferation and differentiation.

Thanks for raising this question, which we did not examine in our initial studies. We agree that TGF β 1/TRAF3 could regulate osteoblast proliferation and differentiation via classical Smad signaling. Indeed, we found that the basal levels of phospho-Smad2 and phospho-Smad3 were higher in BDMPCs from cKO mice than from WT cells. We confirmed that TGF β 1 treatment quickly (within 30') increased the levels of phosphorylated Smad2 and Smad3 in WT cells, but they did not increase in the cKO cells until 120'. We have included these new data in Supplemental Fig. 3. They suggest that TGF β canonical signaling also involves TRAF, which we show gets recruited to the TGF β receptors and discuss how TRAF3 regulates OB differentiation via NF- κ B interactions with the Smad pathway in page 19 of the revised manuscript. Full investigation of these interactions will require significant additional study, which will require additional time and which we will publish in a separate paper.

4) Although the authors showed that TGF-beta inhibited osteoblast differentiation via the activation of Akt, the previous study, by using Akt-deficient mice, clearly demonstrated that Akt enhanced osteoblast differentiation by activating Runx2 transcriptional activity and inhibited apoptosis (Kawamura N et al., PLOS ONE, 2007). The authors need to explain these discrepancies.

We agree that the link between Akt and TGF β 1/TRAF3 to regulate β -catenin during osteoblast differentiation is not supported by our original findings and, as stated above in the response to question 7 from reviewer 1, we have removed these data from the paper.

5) It was reported that osteocyte-intrinsic TGF-beta signaling induces genes involved in perilacunar/canalicular remodeling (Dole NS, et al., Cell Rep, 2018). To examine the involvement of TRAF3 in TGF-beta signaling in osteocytes, the authors should analyze the expression of genes involved in perilacunar/canalicular remodeling, along with RANKL/OPG expression in osteocytes, since TRAF3 is also deficient in osteocytes of Prx1-Cre Traf3 cKO mice.

To address this point, we analyzed the mRNA expression levels of Acp5, Mmp2, Mmp13, Mmp14 and Cathepsin K, genes reported to regulate osteocyte functions and perilacunar/canalicular remodeling, in cortical bone from 12-m-old WT and TRAF3 cKO mice. We found no differences

in the levels of expression of these genes in bone samples from these mice or in the osteocyte lacunar area or the numbers of MMP13+ or CTSK+ osteocytes in the cortical bone of cKO and WT mice (Supplemental Fig. 4). We showed that the cortical bone containing osteocytes from 9-month-old cKO mice has increased expression levels of both RANKL and OPG, but that the RANKL/OPG ratio was higher in the cKO samples than in those of their WT littermates in the original Fig. 5b. The increased levels of active TGFβ1 in the bones of the cKO mice did not result in a change in their perilacunar remodeling. Of course, this is different from the model reported by Dole et al in which deletion of the TGFβR in osteocytes resulted in reduced perilacunar remodeling¹⁰. They did not examine the effects of over-expression of TGFβ in osteocytes on perilacunar remodeling. We think it is unlikely that the TGFβ1 that is activated during the enhanced resorption of TRAF3 cKO mice would go back into the bone matrix through canaliculi to regulate osteocytes, but investigation of that would require significant additional work. We have reported our additional findings on page 15 of the revised manuscript.

6) In Fig. 3g, AT406, an antagonist of IAP, completely abrogated the binding of TRAF3 to TGF-beta receptor 1. In contrast, AT406 treatment only partially restores osteoblast differentiation as shown in Fig. 3h. This indicated that IAP-independent pathway may be involved in TGF-beta induced inhibition of osteoblast differentiation. The authors should discuss TRAF3-independent pathway during TGF-induced inhibition of osteoblast differentiation.

We did comment that TGFβ1 inhibits osteoblast differentiation via degradation of β-catenin in TRAF3-dependent and -independent manners in the original manuscript (now page 12 in the revised version). We thank the reviewer for highlighting this additional evidence that TGFβ1 regulates OB differentiation through TRAF3-independent mechanisms. We have added an additional comment in page 10 of the revised manuscript.

Minor points

1) Generally speaking, osteoclast number and activity in mice is reduced with age due to aging-associated low turnover of bone remodeling. The authors need to explain why there were no differences in osteoclast number, surface and serum TRAcP5b level between young and aged WT mice in Fig. 1h.

We agree that osteoclast number and activity in mice is generally reduced with age due to aging-associated low bone turnover. Bone turnover changes with aging are complex and different from postmenopausal osteoporosis in which there is transiently accelerated bone turnover. Although bone turnover markers are low in aged mice and humans, we found no significant differences in osteoclast number, surface and serum TRACP5b among 3-, 9- and 15- month-old WT mice in our studies (Fig.1), as the Reviewer points out. This may be because 15-month-old mice are still not sufficiently aged and are equivalent to ~50 year old humans who tend to still have normal bone turnover marker levels¹¹. Another possibility is that the genetic background could have an effect here since our WT littermates are actually C57BL6/TRAF3 floxed mice. We have added a comment on this on page 7.

2) The authors described that TRAF3 overexpression prevented TGF-beta-induced beta-catenin phosphorylation. However, in Fig. 4f, the data clearly showed that TGF-beta markedly induced

beta-catenin phosphorylation without activation of Akt in TRAF3 ectopically expressed cells. This indicates that TGF-beta-induced Akt-independent pathway may have a major influence on the regulation of beta-catenin. The authors need to explain these points.

We agree with the reviewer and deleted the AKT data in Fig.4, as discussed in question 4 and in Reviewer-1's question 7.

3) In Fig. 5d and 5e, RelA and RelB localized nucleus and recruited to RANKL promoter without TGF-beta stimulation even in WT cells. The authors should discuss how NF-kappaB signaling was activated in mesenchymal precursor cells in steady state.

Thanks for raising this question. A low level of NF- κ B signaling is required to maintain cell survival even in static conditions. Genetic deletion of RelA is lethal due to massive necrosis of liver cells¹², indicating that a basal level of RelA activity is required to prevent TNF-induced apoptosis of hepatocytes. 'Unstimulated' cultured MPCs have low levels of nuclear RelA and RelB unless the cells are challenged, for example by starvation, which leads to loss of nuclear RelA and RelB. We did not starve cells in these experiments, and thus low levels of nuclear RelA and RelB as well as their binding to the RANKL promoter were detected in the PBS-treated cells. We added a comment on this on pages 14-15.

4) In Fig. 4e, the authors performed IP-western assay by using anti-PI3K antibody. However, there were no descriptions of the antibody against PI3K and immunoprecipitation assay in Method section.

Thanks for raising this issue. The PI3K/AKT data were deleted in the revised version as discussed.

References

1. Lin WW, Hostager BS, Bishop GA. TRAF3, ubiquitination, and B-lymphocyte regulation. *Immunological reviews* 2015;266:46-55.
2. Miao Y, Wu J, Abraham SN. Ubiquitination of Innate Immune Regulator TRAF3 Orchestrates Expulsion of Intracellular Bacteria by Exocyst Complex. *Immunity* 2016;45:94-105.
3. Tseng PH, Matsuzawa A, Zhang W, Mino T, Vignali DA, Karin M. Different modes of ubiquitination of the adaptor TRAF3 selectively activate the expression of type I interferons and proinflammatory cytokines. *Nature immunology* 2010;11:70-5.
4. Mao AP, Li S, Zhong B, et al. Virus-triggered ubiquitination of TRAF3/6 by cIAP1/2 is essential for induction of interferon-beta (IFN-beta) and cellular antiviral response. *The Journal of biological chemistry* 2010;285:9470-6.
5. Hu H, Brittain GC, Chang JH, et al. OTUD7B controls non-canonical NF-kappaB activation through deubiquitination of TRAF3. *Nature* 2013;494:371-4.
6. Cai Q, Sun H, Peng Y, et al. A potent and orally active antagonist (SM-406/AT-406) of multiple inhibitor of apoptosis proteins (IAPs) in clinical development for cancer treatment. *Journal of medicinal chemistry* 2011;54:2714-26.

7. Zhang T, Li Y, Zou P, et al. Physiologically based pharmacokinetic and pharmacodynamic modeling of an antagonist (SM-406/AT-406) of multiple inhibitor of apoptosis proteins (IAPs) in a mouse xenograft model of human breast cancer. *Biopharm Drug Dispos* 2013;34:348-59.
8. Mu Y, Sundar R, Thakur N, et al. TRAF6 ubiquitinates TGFbeta type I receptor to promote its cleavage and nuclear translocation in cancer. *Nature communications* 2011;2:330.
9. Jope RS, Johnson GV. The glamour and gloom of glycogen synthase kinase-3. *Trends in biochemical sciences* 2004;29:95-102.
10. Dole NS, Mazur CM, Acevedo C, et al. Osteocyte-Intrinsic TGF-beta Signaling Regulates Bone Quality through Perilacunar/Canalicular Remodeling. *Cell reports* 2017;21:2585-96.
11. Dutta S, Sengupta P. Men and mice: Relating their ages. *Life sciences* 2016;152:244-8.
12. Beg AA, Sha WC, Bronson RT, Ghosh S, Baltimore D. Embryonic lethality and liver degeneration in mice lacking the RelA component of NF-kappa B. *Nature* 1995;376:167-70.

Reviewers' Comments:

Reviewer #2:

Remarks to the Author:

The authors have satisfactorily addressed the concerns I raised.

Reviewer #3:

Remarks to the Author:

The authors breed TRAF3(flox/flox) mice with a Prx1-Cre strain to specifically disrupt TRAF3 gene expression in mesenchymal progenitor cells (MPCs) to evaluate the role of TRAF3 in bone development and homeostasis. The authors find that the TRAF3-deficient mice develop a form of osteoporosis with age, suggesting that TRAF3 is important in bone homeostasis. The authors propose that TRAF3 is involved in regulating TGF β -induced GSK-3 β and NF- κ B activity in osteoblast precursors. Together, these effects limit the differentiation of osteoblasts and promote the activation of osteoclasts. Based on their observations, the authors suggest some sort of TRAF3-based treatment of osteoporosis.

In Figure 1, the authors present evidence that aged TRAF3-deficient mice develop more pronounced osteoporosis than do control mice. Knockout mice show decreased mineralizing surfaces (MS/BS), mineral apposition rates (MAR), and bone formation rates (BFR) at 9 months of age. These parameters become similar between knockout and wild-type mice at 15 months. Markers of bone resorption are also increased in aged (>9mo) TRAF3-deficient mice. Note: the "red" arrows in 1f and 1h are yellow.

Figure 2 shows that the in vitro differentiation of osteoblasts from TRAF3-deficient bone marrow or bone-derived MPCs (BdMPCs) is decreased. The growth rate of BdMPCs from TRAF3-deficient mice is enhanced, as one might expect with decreased differentiation. The authors propose that TGF β 1, which is known to inhibit osteoblast differentiation, contributes to the decrease in knockout cell differentiation. While the authors confirm that TGF β 1 does have a marked inhibitory effect on BdMPC differentiation, the effect is evident whether WT or knockout cells are used in the experiment. The authors show that mRNA levels for TGF β 1 are similar in the WT and knockout BdMPC in young and old mice (supplemental fig. 1), suggesting that TGF β 1 is not a key factor in preventing the differentiation of cells from older mice. Nevertheless, the authors continue the manuscript with a variety of TGF β experiments.

TRAF3 protein expression in bone and bone marrow is presented for young and old mice and humans. TRAF3 levels decrease with age. Panels 3a and b show Western blot results, while panel c shows histology from young and old mice. While panel c appears to agree with the authors' hypothesis (less TRAF3 in osteoblasts of old mice), there are random triangles pointing to "something". A triangle has three points. It isn't clear which point to follow, and some triangles appear to point to nothing. There are also some dotted lines in the upper right that seem to outline nothing in particular. In Fig. 3e, the authors return to TGF β to test if it might induce loss of TRAF3 in MPC, which it appears to do, in contrast to a few other factors (like TNF) that the authors thought might be relevant. The authors present evidence that there is more active TGF β 1 in bone marrow of old mice than young. Furthermore, the authors propose that this TGF β 1 induces ubiquitination and proteasomal or lysosomal degradation of TRAF3. Fig. 3i shows a TRAF3 IP and ubiquitin Western. As mentioned by other reviewers, the left panel is uninformative (and should be deleted): material appearing on the blot may simply be TRAF3-associated ubiquitinated proteins. The right panel is better (a TRAF3 IP instead of ubiquitin), although it would be helpful to show the molecular weight markers. Presumably, the arrows next to blot are pointing to ubiquitinated TRAF3, but it would be good to know if both bands are of a higher MW than unmodified TRAF3. The remainder of the figure is devoted to showing evidence of recruitment of TRAF3 and ubiquitin ligases to the TGF

receptor, and to providing evidence that perhaps the majority of TGF-induced TRAF3 degradation occurs in lysosomes. Figures 3o-p imply that by inhibiting lysosomal degradation of TRAF3, the inhibitory effects of TGFB1 on differentiation can be mitigated. However, only WT cells are tested in this experiment. I suspect that the authors would see the same effect if TRAF3-deficient cells are used, as the differentiation of TRAF3-deficient cells is also inhibited by TGFB1 (Fig. 2h). Nevertheless, the authors state that, "Importantly, chloroquine inhibited TGF β 1-induced TRAF3 degradation (Fig. 3o) and dose-dependently prevented the inhibition of osteoblast differentiation induced by TGF β 1 (Fig. 3p)". This statement implies that the data support a direct link between TRAF3 degradation and the effects of TGFB, which is not the case.

In Fig. 4, the authors suggest a link between TRAF3, B-catenin, TGFB1 and MPC differentiation. B-cat is known to promote differentiation of MPC, and the authors demonstrate lower levels of B-cat in TRAF3-deficient bone. The authors also show that TGF treatment reduces TRAF3 and B-cat levels in MPC. The authors demonstrate that TGF treatment reduces B-cat expression in both WT and TRAF3-deficient cells, acknowledging that TGF can also induce B-cat degradation through a TRAF3-independent mechanism. In the same panels (c, d), the authors show that reconstitution of TRAF3-deficient cells does not restore B-cat levels (in the absence of TGF), which would have been expected based on the author's model. TRAF3 reconstitution does appear to mitigate TGFB-induced B-cat loss.

To explain the reduced B-cat levels in knockout bone, the authors investigated the activity level of GSK3B, and found more active enzyme (but not more total enzyme) in knockout bone. Overexpression of TRAF3 in pre-OB cells (WT) appears to lower GSK3B activity and turnover of B-cat. Incongruously, a GSK3B inhibitor reversed the inhibitory effects of TGF in MPC from WT mice (Fig. 4j) but had very little effect in cells from TRAF3-deficient mice. Presumably, from the author's model (Fig. 7), a GSK3 inhibitor should substitute for the inhibitory action of TRAF3. This suggests a flaw in the model.

In supplementary figure 3, the authors attempt to evaluate a possible role for TRAF3 in regulating the expression of SMAD proteins. SMADs are involved in mediating TGF receptor signals. While there appears to be no role of TRAF3 in SMAD expression, there may be some role in SMAD phosphorylation, which the authors do not further explore.

In figure 5, the authors explore a possible role of TRAF3 in regulating RANKL production by osteoblasts. RANKL has the potential to activate osteoclasts. If over-produced by TRAF3-deficient OB, it could contribute to osteoporosis in older animals. Experiments in figure 5 provide reasonable evidence that TRAF3-deficient OB produce more RANKL than WT, and that this increased production is likely due to dysregulation of NF-kB2. Note: it's not clear what panel j is supposed to show—it seems to be a quality control test of the QPCR primers used in h and i. There is also no "j" in the legend. Figure 6 is a confirmation/continuation of the experiments in Figure 5.

Supplemental Fig. 4 evaluates markers of perilacunar/canalicular remodeling in WT and KO animals. There is no difference between WT and KO mice. It's not entirely clear what the goal of these experiments was. What was the expected result? Discussion?

In summary, the experiments presented support a role for TRAF3 in preventing osteoporosis. The data also support a role of TRAF3 in modulating RANKL production by OB or OB precursors by regulating NF-kB2. The authors also demonstrate recruitment of TRAF3 to the TGF receptor, and subsequent degradation of TRAF3. However, support for some of the other ideas suggested in the manuscript is less solid.

The authors seem to suggest that lysosomal degradation of TRAF3 is directly related to the inhibition of OB differentiation mediated by TGF. The authors spend a great deal of time evaluating the effects of TGF on OB differentiation, yet the results in Fig. 2c and supplemental fig. 1 show

that the differentiation defect in older TRAF3-deficient cells likely is not driven by TGF. Further, TGF can inhibit OB differentiation whether TRAF3 is there or not (Fig. 2h). The authors imply that treating cells with a lysosomal inhibitor (to stop TRAF3 degradation) is sufficient to mitigate the effects of TGF on differentiation. However, they failed to show what effect the lysosomal inhibitor has on TRAF3-deficient cells. The authors also suggest that TRAF3 inhibits the activity of GSK3B, and that this enhances the B-cat activity necessary for OB differentiation. However, a TRAF3 retrovirus does not restore the expression of B-cat in knockout cells, nor does it change the level of B-cat in WT (Fig. 4d). Furthermore, a GSK3B inhibitor only weakly restores the TGF inhibition of OB differentiation (Fig. 4j). Together, these inconsistencies suggest that the model presented in Fig. 7 is incomplete.

We thank the reviewer for the careful and helpful review of our revised manuscript and Figures. We have addressed these below in italicized text. We have made changes in the manuscript in red text, leaving the original changes in blue so that the previous and new changes are easily identified.

Reviewer #3 (Remarks to the Author):

The authors breed TRAF3(flox/flox) mice with a Prx1-Cre strain to specifically disrupt TRAF3 gene expression in mesenchymal progenitor cells (MPCs) to evaluate the role of TRAF3 in bone development and homeostasis. The authors find that the TRAF3-deficient mice develop a form of osteoporosis with age, suggesting that TRAF3 is important in bone homeostasis. The authors propose that TRAF3 is involved in regulating TGFb-induced GSK-3b and NF-kB2 activity in osteoblast precursors. Together, these effects limit the differentiation of osteoblasts and promote the activation of osteoclasts. Based on their observations, the authors suggest some sort of TRAF3-based treatment of osteoporosis.

In Figure 1, the authors present evidence that aged TRAF3-deficient mice develop more pronounced osteoporosis than do control mice. Knockout mice show decreased mineralizing surfaces (MS/BS), mineral apposition rates (MAR), and bone formation rates (BFR) at 9 months of age. These parameters become similar between knockout and wild-type mice at 15 months. Markers of bone resorption are also increased in aged (>9mo) TRAF3-deficient mice. Note: the “red” arrows in 1f and 1h are yellow.

Thanks for spotting this. We have corrected this error by changing ‘red’ to ‘yellow’ in the legend for Figure 1.

Figure 2 shows that the in vitro differentiation of osteoblasts from TRAF3-deficient bone marrow or bone-derived MPCs (BdMPCs) is decreased. The growth rate of BdMPCs from TRAF3-deficient mice is enhanced, as one might expect with decreased differentiation. The authors propose that TGFB1, which is known to inhibit osteoblast differentiation, contributes to the decrease in knockout cell differentiation. While the authors confirm that TGFB1 does have a marked inhibitory effect on BdMPC differentiation, the effect is evident whether WT or knockout cells are used in the experiment. The authors show that mRNA levels for TGFB1 are similar in the WT and knockout BdMPC in young and old mice (supplemental fig. 1), suggesting that TGFB1 is not a key factor in preventing the differentiation of cells from older mice. Nevertheless, the authors continue the manuscript with a variety of TGFB experiments.

We are not proposing that there is any change in TGFβ1 transcription in these mice, as Suppl. Fig. 1 illustrates and shows that cKO BdMPCs from young and old mice express TGFβ1 at the same level as WT cells. In contrast, our hypothesis, based on our data, is that TGFβ1

protein is released from bone matrix during resorption and is activated under the acid environment generated by osteoclasts in higher amounts in older mice and in the TRAF3 cKO mice as a result of the increased bone resorption in the older WT mice and in the younger cKO mice (Figs. 2g, 3g and 3h). It is this released TGFβ1 that affects the MSCs. As we state in the Discussion, “we propose that the increased osteoblast precursor proliferation in young cKO mice maintains a sufficient pool of cells to differentiate into osteoblasts and maintain bone mass, but increased release of TGFβ1 as the mice age limits differentiation of osteoblast precursors.” In fact, the levels of TGFβ1 in human and mouse serum are considerably elevated, ranging from 20-50 ng/ml¹⁻³. However, serum TGFβ1 exists mainly in its latent form, which does not have biological activity until it is activated in acid environments, such as in resorption lacunae in bone and in chronic kidney diseases where it can result in renal fibrosis⁴⁻⁶. In contrast, only a very low dose (0.3 ng/ml or lower) of active TGFβ1 is needed to inhibit OB differentiation (Fig. 2). Although the active form of TGFβ1 is increased slightly in serum of aged mice and younger cKO mice as well as in the bone of aged human, we believe that this is enough to inhibit bone formation in the bone marrow with aging. Of note, a large body of evidence shows the bone anabolic effect of TGFβ1 inhibition⁷⁻⁹.

TRAF3 protein expression in bone and bone marrow is presented for young and old mice and humans. TRAF3 levels decrease with age. Panels 3a and b show Western blot results, while panel c shows histology from young and old mice. While panel c appears to agree with the authors' hypothesis (less TRAF3 in osteoblasts of old mice), there are random triangles pointing to “something”. A triangle has three points. It isn't clear which point to follow, and some triangles appear to point to nothing. There are also some dotted lines in the upper right that seem to outline nothing in particular.

Thank you for pointing out these errors in Fig. 3c. This appears to have been a Mac/PC switch error that we didn't pick up before resubmission of the Figures. We have replaced the triangles with arrows and ensured that the dotted lines are where they should be.

In Fig. 3e, the authors return to TGFB to test if it might induce loss of TRAF3 in MPC, which it appears to do, in contrast to a few other factors (like TNF) that the authors thought might be relevant. The authors present evidence that there is more active TGFB1 in bone marrow of old mice than young. Furthermore, the authors propose that this TGFB1 induces ubiquitination and proteasomal or lysosomal degradation of TRAF3. Fig. 3i shows a TRAF3 IP and ubiquitin Western. As mentioned by other reviewers, the left panel is uninformative (and should be deleted): material appearing on the blot may simply be TRAF3-associated ubiquitinated proteins.

We have removed the left-hand panel, as requested.

The right panel is better (a TRAF3 IP instead of ubiquitin), although it would be helpful to show the molecular weight markers.

We have added molecular weight markers, as requested.

Presumably, the arrows next to blot are pointing to ubiquitinated TRAF3, but it would be good to know if both bands are of a higher MW than unmodified TRAF3.

We used an anti-Ub antibody to pull down the proteins followed by Western blot analysis of TRAF3. Thus, both bands are ubiquitinated TRAF3, as the reviewer presumed. The molecular weight of unmodified TRAF3 is about 68 kD, while the molecular weight of the lowest band of TRAF3, pulled down by the anti-Ub antibody, as shown in the Figure, is about 75 kD, which is higher than the unmodified TRAF3. The molecular weight of ubiquitin is about 8 kD. Thus, based on the position of these bands in the revised Figure, we believe that the lowest band of TRAF3 (about 75 kD) is mono-ubiquitinated TRAF3, and the middle bands and upper smear signals are poly-ubiquitinated TRAF3. As we described in the Methods section, ubiquitin-recycling was effectively blocked by adding 1 $\mu\text{g/ml}$ ubiquitin aldehyde (Enzo Life Sciences) and other reagents in all experiments studying ubiquitination.

The remainder of the figure is devoted to showing evidence of recruitment of TRAF3 and ubiquitin ligases to the TGF receptor, and to providing evidence that perhaps the majority of TGF-induced TRAF3 degradation occurs in lysosomes. Figures 3o-p imply that by inhibiting lysosomal degradation of TRAF3, the inhibitory effects of TGFB1 on differentiation can be mitigated. However, only WT cells are tested in this experiment. I suspect that the authors would see the same effect if TRAF3-deficient cells are used, as the differentiation of TRAF3-deficient cells is also inhibited by TGFB1 (Fig. 2h). Nevertheless, the authors state that, “Importantly, chloroquine inhibited TGF β 1-induced TRAF3 degradation (Fig. 3o) and dose-dependently prevented the inhibition of osteoblast differentiation induced by TGF β 1 (Fig. 3p)”. This statement implies that the data support a direct link between TRAF3 degradation and the effects of TGFB, which is not the case.

We have now examined the differentiation of TRAF3-deficient BdMPCs into osteoblasts in response to TGF β 1 side by side with WT cells and found, as the reviewer suggested, that the inhibitory effects of TGF β 1 in the cKO cells were slightly mitigated by CQ. However, this occurred at much higher doses than in WT cells, suggesting that TGF β 1 inhibition of OB differentiation is at least partly through lysosomal degradation of TRAF3. Thus, it is possible that, in addition to TRAF3-mediated signaling, other as yet unidentified pathways also mediate the inhibitory effects of TGF β 1 on osteoblast differentiation. As we show in Fig. 5d, nuclear translocation of NF- κ B RelA and RelB was higher in cKO BdMPCs than in WT cells, but this increased only slightly in the cKO cells in response to TGF β 1 treatment, suggesting other pathways might be involved in this process.

We changed the statement to, “Importantly, chloroquine inhibited TGF β 1-induced TRAF3 degradation (Fig. 3o) and dose-dependently prevented the inhibition of osteoblast differentiation induced by TGF β 1 (Fig. 3p) in WT cells. Chloroquine also reduced the inhibition of osteoblast

differentiation induced by TGFβ1 in cKO cells, but only at the highest concentration tested (3,000nM), which supports lysosomal degradation of TRAF3 being a major, but not the only mechanism whereby TGFβ1 inhibits OB formation”.

In Fig. 4, the authors suggest a link between TRAF3, B-catenin, TGFβ1 and MPC differentiation. B-cat is known to promote differentiation of MPC, and the authors demonstrate lower levels of B-cat in TRAF3-deficient bone. The authors also show that TGF treatment reduces TRAF3 and B-cat levels in MPC. The authors demonstrate that TGF treatment reduces B-cat expression in both WT and TRAF3-deficient cells, acknowledging that TGF can also induce B-cat degradation through a TRAF3-independent mechanism. In the same panels (c, d), the authors show that reconstitution of TRAF3-deficient cells does not restore B-cat levels (in the absence of TGF), which would have been expected based on the author’s model. TRAF3 reconstitution does appear to mitigate TGFβ-induced B-cat loss.

Thanks for raising these questions. Over-expression of TRAF3 alone did not increase nuclear β-catenin in either WT or TRAF3 cKO BdmPCs, but it markedly blocked TGFβ1 inhibition of β-catenin nuclear translocation, suggesting that TRAF3 alone does not stimulate OB differentiation via β-catenin, but it does limit the reduced OB differentiation caused by TGFβ-induced degradation of β-catenin. This is consistent with the phenotype of TRAF3 cKO mice, which have accelerated age-related bone loss. Although the younger cKO mice do not have reduced bone mass and their MSCs do not have reduced OB differentiation, these cKO MSCs produce more RANKL, which enhances osteoclast formation and bone resorption, resulting in the release of more active TGFβ1 from bone matrix to inhibit bone formation, which ultimately leads to low bone volume in older cKO mice, as we proposed.

We believe that a TRAF3-independent pathway might also contribute to this β-catenin reduction in cKO cells, which is supported by the data in Fig. 4d. We agree that we did not see full restoration of β-catenin in nuclei of cKO BdmPCs with TRAF3 reconstitution (in the absence of TGFβ), which was confirmed by testing total β-catenin levels in protein lysates of BdmPCs with TRAF3 reconstitution. We were not surprised at this finding since these unstimulated cells have constitutively activated GSK-3β⁰ to degrade β-catenin, reflecting that over-expression of TRAF3 has a limited role to stimulate OB differentiation, but it protects against age-related bone loss, induced by cytokines, such as TGFβ1, because over-expression of TRAF3 significantly reduces TGFβ1-induced inhibition of nuclear β-catenin and phos-GSK3β Ty216.

We have added the following statement to address this issue on page 12, “However, over-expression of TRAF3 in unstimulated cKO cells did not restore the degree of β-catenin translocation to that observed in unstimulated WT cells. We speculate that this reflects the low

level of TGF β expression in these cells and the fact that TRAF3 itself does not possess signaling activity. The inhibitory effects of TRAF3 are observed when it is degraded in response to stimulation of a signaling pathway by a ligand, such as TGF β .”

To explain the reduced B-cat levels in knockout bone, the authors investigated the activity level of GSK3B, and found more active enzyme (but not more total enzyme) in knockout bone. Overexpression of TRAF3 in pre-OB cells (WT) appears to lower GSK3B activity and turnover of B-cat. Incongruously, a GSK3B inhibitor reversed the inhibitory effects of TGF in MPC from WT mice (Fig. 4j) but had very little effect in cells from TRAF3-deficient mice. Presumably, from the author's model (Fig. 7), a GSK3 inhibitor should substitute for the inhibitory action of TRAF3. This suggests a flaw in the model.

We agree that the data we presented appear to be incongruous with our proposal and that the GSK3 β inhibitor should substitute for the inhibitory action of TRAF3. We thank the reviewer for pointing out this deficiency in the data we presented. To examine this issue further, we repeated these experiments using a fuller dose-response range of the inhibitor because the cKO cells have constitutively higher levels of phosphorylated GSK3 β (Ty216), an active form of GSK3 β , as illustrated in Fig. 4f. In the revised figure, we now show that the cKO cells, like the WT cells, do have an increase in differentiation in response to the GSK3 β inhibitor in the presence of TGF β 1, but at higher concentrations (Fig. 4j&k), accompanied by a reduction in p-GSK3 β 1 (Ty216) protein levels (Fig. 4l&m). We have repeated this experiment 3 times with similar results.

We have added a comment on these new findings on page 13 of the revised manuscript, “In addition, the GSK3 β inhibitor, SB-216763, prevented TGF β 1-induced inhibition of OB differentiation from WT BdMPCs (Fig. 4j&k), and reduced the increase in protein levels of p-GSK3 β (Tyr216) (Fig. 4l&m). The inhibitor also prevented TGF β 1-induced inhibition of OB differentiation and the increase in protein levels of p-GSK3 β (Tyr216) from cKO cells (Fig. 4o&p), but this required higher concentrations of the inhibitor, reflecting the higher level of p-GSK3 β (Tyr216) in these cells (Fig. 4l&m). In addition, over-expression of TRAF3 in WT MPCs increased the area of ALP⁺ cells (Fig. 4n&o). “

In supplementary figure 3, the authors attempt to evaluate a possible role for TRAF3 in regulating the expression of SMAD proteins. SMADs are involved in mediating TGF receptor signals. While there appears to be no role of TRAF3 in SMAD expression, there may be some role in SMAD phosphorylation, which the authors do not further explore.

We agree that we have not further explored a possible role for TRAF3 in SMAD phosphorylation. This is an additional potential role for TRAF3 to limit TGF β 1-induced inhibition of OB formation, which we will investigate in future studies, since it is beyond the scope of this study. We hope that the reviewer agrees.

In figure 5, the authors explore a possible role of TRAF3 in regulating RANKL production by osteoblasts. RANKL has the potential to activate osteoclasts. If over-produced by TRAF3-deficient OB, it could contribute to osteoporosis in older animals. Experiments in figure 5 provide reasonable evidence that TRAF3-deficient OB produce more RANKL than WT, and that this increased production is likely due to dysregulation of NF- κ B2. Note: it's not clear what panel j is supposed to show—it seems to be a quality control test of the QPCR primers used in h and i. There is also no “j” in the legend. Figure 6 is a confirmation/continuation of the experiments in Figure 5.

We thanks the reviewer for spotting the errors in the legend for Fig 5, which we had changed to better conform to the Nat Commun format, but failed to update the legend for this Figure. We have changed the legend to match the various components of the Figure.

Supplemental Fig. 4 evaluates markers of perilacunar/canalicular remodeling in WT and KO animals. There is no difference between WT and KO mice. It's not entirely clear what the goal of these experiments was. What was the expected result? Discussion?

These are additional experiments that we did at the request of Reviewer 2 to determine if deletion of TRAF3 in MPCs affected these parameters. We have modified to text to make it clearer why we did this additional work, “Thus, the increase in levels of total TGF β 1 in old WT mouse tibiae or in active TGF β 1 in bone marrow in old mice and in adult human bone samples might increase perilacunar remodeling.”

In summary, the experiments presented support a role for TRAF3 in preventing osteoporosis. The data also support a role of TRAF3 in modulating RANKL production by OB or OB precursors by regulating NF- κ B2. The authors also demonstrate recruitment of TRAF3 to the TGF receptor, and subsequent degradation of TRAF3. However, support for some of the other ideas suggested in the manuscript is less solid.

The authors seem to suggest that lysosomal degradation of TRAF3 is directly related to the inhibition of OB differentiation mediated by TGF. The authors spend a great deal of time evaluating the effects of TGF on OB differentiation, yet the results in Fig. 2c and supplemental fig. 1 show that the differentiation defect in older TRAF3-deficient cells likely is not driven by TGF. Further, TGF can inhibit OB differentiation whether TRAF3 is there or not (Fig. 2h). The authors imply that treating cells with a lysosomal inhibitor (to stop TRAF3 degradation) is sufficient to mitigate the effects of TGF on differentiation. However, they failed to show what effect the lysosomal inhibitor has on TRAF3-deficient cells. The authors also suggest that TRAF3 inhibits the activity of GSK3B, and that this enhances the B-cat activity necessary for OB differentiation. However, a TRAF3 retrovirus does not restore the expression of B-cat in knockout cells, nor does it change the level of B-cat in WT (Fig. 4d). Furthermore, a GSK3B

inhibitor only weakly restores the TGF inhibition of OB differentiation (Fig. 4j). Together, these inconsistencies suggest that the model presented in Fig. 7 is incomplete.

We hope that our responses above have sufficiently addressed all these concerns and we thank the reviewer for his/her helpful criticisms and suggestions, which have helped us to significantly improve the clarity of our story. The young TRAF3 cKO mice do not have a bone phenotype and young cKO MPCs do not have impaired OB differentiation either, suggesting that the bone phenotype of older cKO mice is driven by a secondary factor(s). We found that the active form of TGF β 1 is increased in older cKO mice. We brought TGF β 1 into the story here based not only on the TRAF3 cKO mouse model, but also on the “normal” bone aging process, in which TGF β 1 induces TRAF3 ubiquitination and lysosomal degradation, which mimics the changes in the adult TRAF3 cKO mice, at least in part. After TRAF3 degradation (or with TRAF3 cKO), TGF β 1 inhibits OB differentiation in a TRAF3-dependent manner, evidenced by our finding that the cKO cells are more sensitive to TGF β 1 treatment as a consequence of the constitutively activated GSK3 β Ty216, downstream from TRAF3, and in a TRAF3-independent manner, as now noted in page 13, and as the reviewer suggests that “TGF β 1 can inhibit OB differentiation whether TRAF3 is there or not”. As the reviewer suggested, we have added the data that the lysosomal inhibitor less effectively rescues TGF β 1 inhibition of OB differentiation in Fig. 3p, as discussed above. “TRAF3 retrovirus does not restore the expression of B-cat in knockout cells, nor does it change the level of B-cat in WT (Fig. 4d)” probably because GSK-3 β is constitutively activated to degrade β -catenin in unstimulated cKO cells, reflecting that over-expression of TRAF3 does not stimulate OB differentiation via β -catenin. However, TRAF3 protects against age-related bone loss by limiting degradation of β -catenin by cytokines, such as TGF β 1, because over-expression of TRAF3 significantly inhibits TGF β 1-induced reduction of nuclear β -catenin and phos-GSK3 β Ty216.

We think that the GSK3 β inhibitor restores the TGF β 1-induced inhibition of OB differentiation less effectively in the cKO cells than in WT cells (Fig. 4j) because cKO cells have increased phos-GSK3 β Ty216 and need higher doses of the inhibitor to block it, as we show in the new Fig. 4j&k. We modified the model in Fig.7 and we hope that the reviewer will be satisfied with our revised model.

References

1. Redondo S, Navarro-Dorado J, Ramajo M, et al. Age-dependent defective TGF-beta1 signaling in patients undergoing coronary artery bypass grafting. *J Cardiothorac Surg* 2014;9:24.
2. Zhang N, Wu XY, Wu XP, et al. Relationship between age-related serum concentrations of TGF-beta1 and TGF-beta2 and those of osteoprotegerin and leptin in native Chinese women. *Clinica chimica acta; international journal of clinical chemistry* 2009;403:63-9.

3. Okamoto Y, Gotoh Y, Uemura O, Tanaka S, Ando T, Nishida M. Age-dependent decrease in serum transforming growth factor (TGF)-beta 1 in healthy Japanese individuals; population study of serum TGF-beta 1 level in Japanese. *Dis Markers* 2005;21:71-4.
4. Kanwar YS. TGF-beta and renal fibrosis: a Pandora's box of surprises. *The American journal of pathology* 2012;181:1147-50.
5. Meng XM, Huang XR, Xiao J, et al. Diverse roles of TGF-beta receptor II in renal fibrosis and inflammation in vivo and in vitro. *J Pathol* 2012;227:175-88.
6. Meng XM, Nikolic-Paterson DJ, Lan HY. TGF-beta: the master regulator of fibrosis. *Nat Rev Nephrol* 2016;12:325-38.
7. Mohammad KS, Chen CG, Balooch G, et al. Pharmacologic inhibition of the TGF-beta type I receptor kinase has anabolic and anti-catabolic effects on bone. *PloS one* 2009;4:e5275.
8. Biswas S, Nyman JS, Alvarez J, et al. Anti-transforming growth factor ss antibody treatment rescues bone loss and prevents breast cancer metastasis to bone. *PloS one* 2011;6:e27090.
9. Ehnert S, Baur J, Schmitt A, et al. TGF-beta1 as possible link between loss of bone mineral density and chronic inflammation. *PloS one* 2010;5:e14073.
10. Doble BW, Woodgett JR. GSK-3: tricks of the trade for a multi-tasking kinase. *Journal of cell science* 2003;116:1175-86.

Reviewers' Comments:

Reviewer #3:

Remarks to the Author:

The authors have adequately addressed my criticisms.

One minor edit: add WT and TRAF3KO labels to figure 4L.